CERN-TH-2024-042

# Hilbert Space Diffusion in Systems with Approximate Symmetries

Rahel L. Baumgartner[1], Luca V. Delacrétaz[2,3], Pranjal Nayak[4] & Julian Sonner[1]

[1]*Department of Theoretical Physics, University of Geneva , 1211 Genève 4, Suisse*
[2]*Kadanoff Center for Theoretical Physics, University of Chicago, Chicago, IL 60637, USA*
[3]*James Franck Institute, University of Chicago, Chicago, IL 60637, USA*
[4]*Department of Theoretical Physics, CERN, 1211 Genève 23, Suisse*

`rahel.baumgartner@unige.ch, lvd@uchicago.edu, pranjal.nayak@cern.ch,`
`julian.sonner@unige.ch`

## Abstract

Random matrix theory (RMT) universality is the defining property of quantum mechanical chaotic systems, and can be probed by observables like the spectral form factor (SFF). In this paper, we describe systematic deviations from RMT behaviour at intermediate time scales in systems with approximate symmetries. At early times, the symmetries allow us to organize the Hilbert space into approximately decoupled sectors, each of which contributes independently to the SFF. At late times, the SFF transitions into the final ramp of the fully mixed chaotic Hamiltonian. For approximate continuous symmetries, the transitional behaviour is governed by a universal process that we call Hilbert space diffusion. The diffusion constant corresponding to this process is related to the relaxation rate of the associated nearly conserved charge. By implementing a chaotic sigma model for Hilbert-space diffusion, we formulate an analytic theory of this process which agrees quantitatively with our numerical results for different examples.

# 1    Introduction

Emergence of universality in various distinct physical systems is one of the most powerful tools that enables us to explore and understand underlying physical properties of complex systems. This feature of physics has shed light on many fundamental aspects of physical systems that would have otherwise not been conspicuous. One of the most important universality principles that helps us understand the spectral properties of strongly-interacting systems is random matrix universality [1]. It has found application in studies of various physical systems like disordered systems, diffusive systems and 2D gravity among others (see [2] for a review on various applications). More recently, it has also shed light on the properties of blackholes and the resolution of the information paradox in gravitational systems on hyperbolic manifolds [3–6]. In reference to physical systems, the averaging over theories (through averaging over their Hamiltonian as modelled by an RMT) should only been seen as a tool to retrieve the universal signals in the physical observations like correlation functions. Conceptually, the averaging over the Hamiltonians facilitates a coarse graining over a part of the physical spectrum that is otherwise hard to scrutinise. The exact nature of the coarse-graining thus facilitated is still an open question and depends on the physical system as well as the observables under consideration. Not all observables are self-averaging or are susceptive to aforementioned averaging. It is important to understand what are such observables and the physical information carried by them about the system of interest.

Random matrix universality is also crucial in our understanding of how closed quantum systems thermalise. Unlike classical systems where ergodicity in phase space is the guiding principle for how systems thermalise, the same can't be said for the quantum mechanical systems. Eigenstate thermalisation hypothesis (ETH) conjectures that pure states in complex quantum systems are indistinguishable from thermal states. The coarse-graining inherent to an RMT description together with ETH provide the quantum mechanical understanding behind thermalisation [7, 8]. Therefore, understanding the emergence of these features in physical systems is important (see [9] for a review). In this work, we study the slowest modes that govern the late time physics in systems with global symmetries. Such modes are manifested through the spectral properties of the spectrum interpreted through the lens of RMT universality, as we now describe. In the context of holography, these slowest modes that describe the RMT behaviour of physical systems also control their late time behaviour and provide the resolution of unitarity [4, 6].

Similarity of the statistical properties of a physical spectrum to those of an RMT is used

as a quantum mechanical definition of chaos [1]. The critical property of an RMT that distinguishes it from integrable systems is the presence of level-repulsion that is contrary to the typically degenerate spectrum in the latter [10]. Degeneracy of the spectrum results from underlying symmetries of the physical system. This is not indifferent from classical integrable theories, where sufficiently many conserved quantities permit for a description in terms of independent action-angle variables. Quantum mechanical integrable systems are also believed to possess large amount of symmetries that are absent in strongly-interacting chaotic systems. A quantity that is often used to test for such RMT behaviour is the spectral form factor (SFF), the Fourier transform to the time domain of the spectral two point correlation function [11, 12]. In RMT, this quantity has a distinct shape described by an initial linear ramp, followed by a plateau phase (see Fig. 1 for a schematic depiction of this behaviour). The presence of a ramp-plateau in the SFF of a generic system is a signature of level-repulsion in the spectral statistics and implies quantum chaos.

Going beyond random matrix universality, a key question for physical quantum systems is how spectral statistics at high energy *deviate* from RMT. It was recognized early on in a single-particle context that this deviation may also exhibit universal features [11, 13]. Intuitively, any structure in a Hamiltonian leads to loss of spectral rigidity beyond some energy scale, usually called the Thouless energy $\Delta E \gtrsim E_{\text{Th}}$. In real time dynamics probed, e.g., through the SFF, this leads to a 'bump' before the Thouless time $t_{\text{Th}} \equiv 1/E_{\text{Th}}$: the linear in time ramp predicted by RMT is approached from above (see, e.g., [14–19]). Characterizing this overshoot of the ramp in realistic quantum many-body systems compared to RMT has been the subject of several recent papers [20–25].

In this work, we are interested in understanding how the presence of approximate symmetries in a physical system affects its chaotic properties. We will see that approximate symmetries lead to a simple yet widespread situation where the overshoot of the ramp can be precisely characterized (Fig. 1). Approximate symmetries split the Hilbert space in approximately decoupled sectors labelled a charge $q = 1, 2, \ldots, Q$. In local systems, the number of sectors $Q$ typically grows polynomially with system size; it can therefore be large but is much smaller than the Hilbert space dimension $N$, and we will be interested in the limit $N \gg Q \gg 1$. We compute the SFF in such systems and show that the slowest modes that govern the deviations from the RMT behaviour are proportional to the symmetry breaking terms in the Hamiltonian. For a system with weakly-broken symmetries, at earlier times the SFF is the same as that of a theory where the symmetry is preserved. The symmetry breaking slow modes are responsible for the transition to the SFF of an RMT at late times (see Fig. 1).

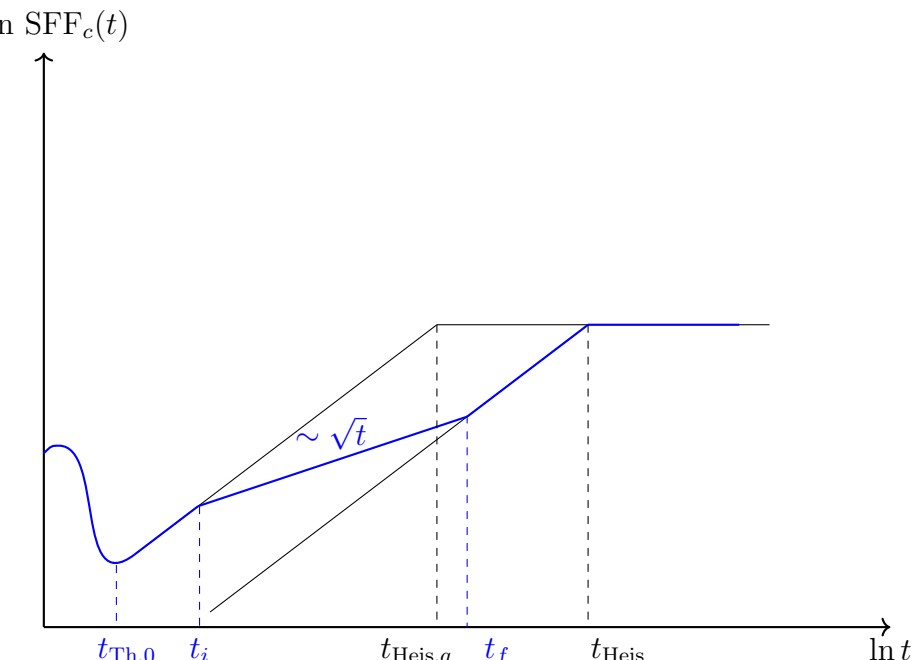

Figure 1: Behaviour of the SFF in systems with approximate symmetries. The 'local' diffusive behaviour occurs between $t_i \simeq 1/(4\pi\Gamma)$ and $t_f \simeq Q^2/(4\pi\Gamma)$, where $Q$ is the number of symmetry sectors of the unperturbed symmetric theory and $\Gamma$ is the 'Hilbert space diffusivity' given in Eq. (1.1). We have marked Thouless time of the symmetric theory, $t_{\mathrm{Th},0}$, at which the ramp begins. The SFF of the symmetric theory plateaus at Heisenberg time, $t_{\mathrm{Heis},q} = Q\,t_{\mathrm{Heis}}$.

When the symmetry-breaking perturbation only couples nearby sectors, this transition is described in the thermodynamic limit as diffusion between various charge sectors in the Hilbert space. We refer to this as *Hilbert space diffusion*. In other words, we demonstrate how the symmetry breaking modes correlate different charge sectors through a process of diffusion. Therefore, even a small amount of symmetry breaking that leads to interactions between only a few charge sectors can give rise to sufficient ergodicity to replicate RMT behaviour at late times. This is again a manifestation of the fact RMT statistics arise even when the underlying theory is not an RMT.

The rate describing this process is given by a two-point function of the charge operator $Q$:

$$\Gamma \simeq \lim_{\omega \to 0} \int dt\, e^{i\omega t} \langle \dot{Q}(t)\dot{Q}\rangle\,. \tag{1.1}$$

When the symmetry is exact, $Q$ is conserved and $\Gamma = 0$. Weakly breaking the symmetry $H_0 \to H_0 + \epsilon V$, non-conservation of the charge $\dot{Q} = i[\epsilon V, Q]$ will lead to a rate $\Gamma \sim \epsilon^2$. This

rate $\Gamma$ is both the inverse timescale marking the onset of Hilbert space diffusion ($t_i = 1/(4\pi\Gamma)$ in Fig. 1), and the diffusivity. We will show that diffusive exploration of the Hilbert space then leads to a $\sqrt{t}$ growth of the SFF, shown in Fig. 1, which eventually terminates when this curve reaches the linear RMT ramp expected for a system without any symmetry. This happens at the time,

$$t_f \simeq \frac{Q^2}{4\pi\Gamma} \ , \tag{1.2}$$

which is the time required for a random walk with diffusivity $\Gamma$ to explore a region of size $Q$. This therefore identifies the Thouless time for systems with approximate symmetries, $t_{\mathrm{Th}} = t_f$.

At the other extreme, we also consider the case where the symmetry is weakly broken by a perturbation that correlates all the different sectors with each other. Such a 'non-local' exploration of Hilbert space results in an exponentially fast transition to a full ergodic behaviour. Interestingly, the competition between the linear growth of the SFF (the ramp) and the exponentially decaying amount of symmetry in such systems can give rise to local extrema in the SFF of such systems.

While similar mechanisms have been considered in a single-body context [13, 26], a key question in many-body quantum chaos is to understand which insights from free particles still apply to interacting systems. Our results only rely on symmetries, and are therefore non-perturbative in the coupling. They apply to any many-body systems, such as spin chains or QFTs, with approximate global symmetries. Another important feature in the many-body context is that each symmetry sector is exponentially large, thereby precluding localization.

**Plan of the paper**

In this paper, we first describe a toy model with $\mathbb{Z}_Q$ symmetry. The Hamiltonian of such a system is a block-diagonal matrix with $Q$ blocks. We consider a model where each of these blocks is governed by an independent RMT. The $\mathbb{Z}_Q$ symmetry is weakly broken by an interaction potential. We consider two cases: (1) 'local' Hilbert-space interactions given by off-diagonal terms in the Hamiltonian that correlate each block with the two adjacent blocks; and, (2) 'non-local' Hilbert-space exploration where the perturbation correlates all the charge sectors with each other through a random matrix of the size of full Hilbert space. In **section 2**, we study the behaviour of the SFF of such systems using the Fermi's Golden rule. These systems are then studied numerically in **section 3**; there we also consider a

more physical system with an approximate U(1) symmetry: the SYK model with charged fermions $c^\dagger, c$, where the U(1) symmetry is broken by adding a small charge-2 perturbation to the Hamiltonian $c^\dagger ccc + \text{h.c.}$. We find excellent quantitative agreement between analytic predictions and numerical evaluations of spectral form factors in these systems. In **section 4**, we provide an independent proof of these results using an effective field theory (EFT) defined as a $\sigma$-model. The $\sigma$-model is described by the pseudo-Goldstone modes, corresponding to the explicit and spontaneous symmetry breaking $\text{U}(1,1|2)^Q \to \text{U}(1|1) \times \text{U}(1|1)$. Lastly, we end our paper with some discussion and comments about the applications of these results and future directions in **section 5**.

# 2  Fermi Golden Rule and Hilbert space diffusion

Symmetries reduce the rigidity of the spectrum of chaotic Hamiltonians, as eigenvalues in different symmetry sectors do not repel. When the symmetry is exact, one usually studies these sectors independently. This cannot be done when the symmetry is instead only approximate, as the dynamics allows for the exploration of all sectors at late times. The loss of spectral rigidity is in this case physical, and depends on the timescale at which one probes the system: the symmetry appears to hold at early times and the systems behaves as if the sectors where decoupled, whereas at very late times all sectors are explored and the system behaves as if there was no symmetry.

We study this quantitatively with a simple example of a system with an approximate $\mathbb{Z}_Q$ symmetry, broken by a small dimensionless parameter $\epsilon \ll 1$: $H = H_0 + \epsilon V$. When $\epsilon = 0$, one can label states by their $\mathbb{Z}_Q$ eigenvalue $q = 1, 2, \ldots, Q$ and define a real time 'partition function' in each sector,

$$Z_q(t) = \frac{1}{N_q} \sum_{i=1}^{N_q} e^{iE_{i,q}t}, \qquad q \in \{1, 2, \ldots, Q\}, \tag{2.1}$$

which we normalized such that $Z_q(0) = 1$.[1] Let us assume for simplicity that all sectors have the same size $N_q$. After the Thouless time $t_{\text{Th},0}$ of the unperturbed Hamiltonian $H_0$ (and

---

[1] When studying physical systems with several energy scales, one may wish to only focus on a micro-canonical window of states around some energy, or work in the canonical ensemble and study statistics as a function of $\beta$. This will not play an important role here.

before its Heisenberg time), each sector will have an SFF described by RMT

$$\langle Z_q(t) Z_{q'}^*(t) \rangle \simeq \frac{\delta_{qq'}}{N_q} \frac{2}{b} \frac{t}{t_{\text{Heis},q}} \,, \tag{2.2}$$

with $b = 1, 2, 4$ for GOE, GUE, GSE statistics respectively, and $t_{\text{Heis},q}$ is the Heisenberg time in the $q$th sector, defined as the average spacing between eigenvalues times $2\pi$ (for a microcanonical window of size $\Delta E$ with constant density of states, $t_{\text{Heis},q} = 2\pi N_q/\Delta E$). The total SFF instead does not have regular RMT behaviour: the ramp is $Q$ times larger due to the decoupled sectors:

$$\text{SFF}(t) \equiv \langle |\tfrac{1}{Q} \sum_{q=1}^{Q} Z_q|^2 \rangle = Q \times \frac{1}{QN_q} \frac{2}{b} \frac{t}{Qt_{\text{Heis},q}} \tag{2.3}$$
$$\equiv Q \times \text{SFF}_{\text{RMT}}(t) \,,$$

where $\text{SFF}_{\text{RMT}}(t)$ denotes the expected SFF ramp of a $QN_q \times QN_q$ matrix. The 'dip-ramp' feature of the SFF is sometimes called 'correlation-hole' whose depth reflects correlations and eigenvalue repulsion. We see here that the correlation-hole is indeed shallower when uncorrelated spectra are superimposed.

Let us now turn on the $\mathbb{Z}_Q$ symmetry breaking term $0 \neq \epsilon \ll 1$. From Fermi's Golden rule type arguments (which are spelled out below), one expects the $\mathbb{Z}_Q$ sectors to mix at a rate $\Gamma \sim \epsilon^2$. At times $t \gtrsim 1/\Gamma$, the factor of $Q$ enhancement in (2.3) is therefore expected to drop, until it eventually reaches 1 at late times, in agreement with RMT. Eq. (2.3) is then replaced by a time-dependent ramp,

$$\text{SFF}(t) = N_{\text{sectors}}(t) \times \text{SFF}_{\text{RMT}}(t) \,. \tag{2.4}$$

Characterizing the overshoot of the ramp in realistic quantum many-body systems compared to RMT has been the subject of several recent papers [20–25,27,28]. In the following, we will establish the form of the function $N_{\text{sectors}}(t)$ in the situation of approximate symmetries. The behaviour of $N_{\text{sectors}}(t)$ will sensitively depend on *how* the symmetry is broken. A physically

relevant situation is the case where the symmetry is broken 'locally' in the Hilbert space:

$$
H = H_0 + \epsilon V\,, \qquad H_0 = \begin{pmatrix} H_0^1 & & & & \\ & H_0^2 & & & \\ & & \ddots & & \\ & & & & H_0^Q \end{pmatrix}, \qquad V = \begin{pmatrix} 0 & V_1 & 0 & & V_Q \\ V_1^\dagger & 0 & V_2 & & \\ 0 & V_2^\dagger & \cdot & \cdot & \\ & & \cdot & \cdot & V_{Q-1} \\ V_Q^\dagger & & & V_{Q-1}^\dagger & 0 \end{pmatrix} \equiv V_+ + V_-\,,
$$

(2.5)

i.e. $V$ only connects sectors of charge $q$ and $q \pm 1$ (we have written $V = V_+ + V_-$ as a sum of terms increasing and decreasing charge, $[Q, V_\pm] = \pm V_\pm$, for future convenience). This situation naturally arises, e.g., in systems with an approximate $U(1)$ symmetry, broken by a charge $q_0 \in \mathbb{Z}$ operator in the Hamiltonian: we will consider such an example in the context of charged SYK in Sec. 3.3. We are considering $q_0 = 1$ for simplicity, but symmetry breaking by any order one charge $q_0$ will lead to similar local exploration of the Hilbert space. We will also consider an alternative at the end of this section, where $V$ is not sparse and instead connects any two sectors, allowing for faster exploration of the whole Hilbert space.

Before we proceed, we would like to mention a particular example of the case when the time reversal symmetry is broken explicitly and the ensemble diffuses from GOE to GUE. This has previously been studied and is referred to as "Dyson's Brownian-motion model" [29–33]. The motivation in these studies was to give an upper bound on time reversal breaking of nuclear physics. Recently, in [34] an example was considered where the Hilbert space is a tensor product of two independent Hilbert spaces. An interaction Hamiltonian correlating these distinct Hilbert spaces induces a diffusive behaviour that looks strikingly similar to what we discuss in our paper.

## 2.1 Hopping rates

Writing the eigenstates of $H_0$ as $|i, q\rangle_0$ such that

$$
H_0|i, q\rangle_0 = |i, q\rangle_0 E_i\,, \qquad \hat{Q}|i, q\rangle_0 = |i, q\rangle_0 q\,,
$$

(2.6)

the perturbed eigenstates are, to linear order in $\epsilon$,

$$
|i, q\rangle = |i, q\rangle_0 + \epsilon \sum_j \sum_{q'=q\pm 1} |j, q'\rangle_0 \frac{{}_0\langle j, q'|V|i, q\rangle_0}{E_j - E_i} + O(\epsilon^2)
$$

(2.7)

The probability for an initial state $|i, q\rangle_0$ to evolve into the state $|j, q+1\rangle_0$ at time $t$ can then be found by introducing a complete basis of perturbed eigenstates above; this gives

$$\left|_0\langle j, q+1|e^{-iHt}|i, q\rangle_0\right|^2 = \epsilon^2 \left|\langle j, q+1|V|i, q\rangle\right|^2 t^2 \left(\frac{\sin\left((E_j - E_i)t/2\right)}{(E_j - E_i)t/2}\right)^2 + O(\epsilon^3). \quad (2.8)$$

To find the total probability of being in the $q+1$ sector, one sums over $j$. We will assume typicality (ETH) and replace the matrix element $|\langle j, q+1|V|i, q\rangle|^2$ by its average:

$$|\langle j, q+1|V|i, q\rangle|^2 \simeq \frac{1}{2\pi\rho(E_j, q)}\langle V_- V_+\rangle_{E_i, q}(\omega). \quad (2.9)$$

Here $\rho(E_j, q)$ is the density of states at energy $E_j$ and charge $q$, and the second factor is the two-point function of $V_+$ and $V_-$ (the parts of $V$ that respectively increase and decrease charge, see (2.5)) at frequency $\omega = E_j - E_i$ in a microcanonical window of energy $E_i$ and charge $q$:

$$\langle V_- V_+\rangle_{E_i, q}(\omega) \equiv \int dt\, e^{i\omega t} \frac{1}{N_w} \sum_{E \simeq E_i} \langle E, q|V_-(t)V_+(0)|E, q\rangle \quad (2.10)$$

($N_w$ is the number of states in the window). Returning to (2.8) and summing over $j$, one finds

$$\sum_j \left|_0\langle j, q+1|e^{-iHt}|i, q\rangle_0\right|^2 \simeq \frac{\epsilon^2 t^2}{2\pi}\int d\omega\, \langle V_- V_+\rangle_{E_i, q}(\omega)\left(\frac{\sin\left(\omega t/2\right)}{\omega t/2}\right)^2$$
$$\simeq t\epsilon^2 \lim_{\omega \to 0}\langle V_- V_+\rangle_{E_i, q}(\omega). \quad (2.11)$$

In the last line, we assumed $t$ to be greater than the Thouless time of the unperturbed Hamiltonian, so that the correlator $\langle VV\rangle_{E_i, q}(\omega)$ for $\omega \lesssim 1/t$ is well approximated by its limit $\omega \to 0$.

The rate to increase charge can be read from (2.11) and is

$$\Gamma_+ = \epsilon^2 \lim_{\omega \to 0}\langle V_- V_+\rangle_{E_i, q}(\omega) \quad (2.12)$$

This rate depends on the energy and charge of the initial state. The rate to decrease charge is given by an identical expression with $V_+ \leftrightarrow V_-$.[2] This result holds for general quantum

---

[2]For quantum systems with an approximate continuous symmetry (say, $U(1)$), these hopping rates are related to (but larger than) the total charge relaxation rate $\Gamma_{\text{relax}}$. This latter rate is given by a Kubo formula $\Gamma_{\text{relax}} = \frac{\epsilon^2}{T\chi_{QQ}}\lim_{\omega \to 0}\langle VV\rangle(\omega)$ [35], where the factor of $T\chi_{QQ}$ accounts for the preferred hopping direction towards lower charge to minimize the free energy.

systems with approximate symmetries. For the simple RMT setting discussed above, $\Gamma_+ = \Gamma_- \equiv \Gamma$ is independent of charge and energy, and (2.12) becomes

$$\Gamma = \frac{t_{\text{Heis}}}{Q}\epsilon^2|V|^2 \tag{2.13}$$

where $|V|^2$ is the average matrix element of $V$, and $t_{\text{Heis}}$ the Heisenberg time of the full $QN_q \times QN_q$ matrix, defined by the total average density of states $\rho$ as $t_{\text{Heis}} \equiv 2\pi\rho$.

## 2.2  Hilbert space diffusion

Asymptotically, a state therefore hops from sectors $q \to q + 1$ and $q \to q - 1$ with rate $\Gamma$. Consider now a general state $|\psi\rangle$ with distribution in the various charge sectors:

$$P(t, q) \equiv |\langle q|\psi(t)\rangle|^2 . \tag{2.14}$$

Because of the hopping between sectors with rate $\Gamma$, this distribution satisfies a differential equation

$$\begin{aligned}
\partial_t P(t, q) &\simeq -2\Gamma P(t, q) + \Gamma \left( P(t, q + 1) + P(t, q - 1) \right) \\
&= \Gamma d_q^2 P(t, q) ,
\end{aligned} \tag{2.15}$$

where we introduced a discrete $q$ derivative $d_q f(q) = f(q + \frac{1}{2}) - f(q - \frac{1}{2})$. $P(t, q)$ therefore satisfies an approximate diffusion equation ('Hilbert space diffusion')[3], with diffusivity $\Gamma$ given by (2.13). Fourier transforming to $P(t, n) = \frac{1}{Q} \sum_{q=0}^{Q-1} e^{i2\pi nq/Q} P(t, q)$ with $n = 0, 1, \ldots, Q - 1$, the solution to Eq. (2.15) with periodic boundary conditions is

$$P(t, n) = e^{-\Gamma_n t} P(0, n) , \qquad \text{with} \qquad \Gamma_n = 4\Gamma \sin^2 \frac{\pi n}{Q} . \tag{2.16}$$

For closed boundary conditions (i.e., $\langle Q|V|1\rangle = 0$) one has instead $\Gamma_n = 4\Gamma \cos^2 \frac{\pi n}{2Q}$ with $n = 1, \ldots, Q$. Note that total probability is conserved, $\Gamma_0 = 0$. The effective number of decoupled sectors at a time $t$ is given by the sum over sectors, times the probability of

---

[3]This diffusive exploration of the Hilbert space was also identified in Ref. [24], who considered a similar toy model as the one studied in this section.

staying within each sector:[4]

$$N_{\text{sectors}}(t) = \sum_{n=0}^{Q-1} e^{-\Gamma_n t}.\tag{2.17}$$

This sum can be readily evaluated using the rates in (2.16). It can also be estimated as follows: at $t \lesssim \frac{1}{\Gamma}$, all sectors are conserved and $N_{\text{sectors}} \simeq Q$. Instead at $t \gtrsim \frac{Q^2}{\Gamma}$, all sectors have decayed except for $n = 0$, and $N_{\text{sectors}} \simeq 1$. Finally, at intermediate times $\frac{1}{\Gamma} \lesssim t \lesssim \frac{Q^2}{\Gamma}$, for $Q \gg 1$ one can approximate the sum in (2.17) with an integral to obtain (for any boundary condition)

$$N_{\text{sectors}}(t) \approx Q \int \frac{dk}{2\pi} e^{-\Gamma k^2 t} = \frac{Q}{\sqrt{4\pi\Gamma t}}.\tag{2.18}$$

This $1/\sqrt{t}$ decay is characteristic of diffusion in one dimension. In summary, the SFF well before the Heisenberg time is expected to be given by

$$\text{SFF}(t) = N_{\text{sectors}}(t)\text{SFF}_{\text{RMT}}(t), \qquad \text{with} \qquad N_{\text{sectors}}(t) \approx \begin{cases} Q & \text{for } t \lesssim \frac{1}{4\pi\Gamma} \\ \frac{Q}{\sqrt{4\pi\Gamma t}} & \text{for } \frac{1}{4\pi\Gamma} \lesssim t \lesssim \frac{Q^2}{4\pi\Gamma} \\ 1 & \text{for } \frac{Q^2}{4\pi\Gamma} \lesssim t. \end{cases}\tag{2.19}$$

This prediction is illustrated in Fig. 1.

## 2.3 'Non-local' exploration of Hilbert space

We now turn to a qualitatively different class of approximate symmetries, where the symmetry breaking perturbation connects all sectors. The local exploration of the Hilbert space is then replaced by non-local exploration, and the effective number of decoupled sectors decays more rapidly with time. As a simple example, consider the RMT from Eq. (2.5) with $\mathbb{Z}_Q$ symmetry broken now by a full $QN_q \times QN_q$ random matrix $V$. The local evolution equation (2.15) is replaced by

$$\partial_t P(t, q) = -Q\Gamma P(t, q) + \Gamma \sum_{q'} P(t, q').\tag{2.20}$$

or, in matrix form: $\partial_t P(t) = -MP(t)$ with the matrix $M_{qq'} = \Gamma(Q\delta_{qq'} - 1)$. The solution is $P(t) = e^{-Mt}P(0)$. $M$ has eigenvalues $\Gamma Q$ (with degeneracy $Q - 1$) and $0$ (with degeneracy

---

[4]This estimate of the effective number of sectors will be fully justified in a $\sigma$-model approach in Sec. 4.

1, the total probability is conserved). The effective number of decoupled sectors is then,

$$N_{\text{sectors}}(t) = 1 + (Q-1)e^{-Q\Gamma t}. \qquad (2.21)$$

This is compared to numerics for RMT in Fig. 3.

# 3 Numerics

In this section, we numerically test the prediction of Hilbert space diffusion (2.19) for several systems with approximate symmetries. We first consider as a toy model the $\mathbb{Z}_Q$-symmetric RMT following (2.5) in which we have good analytical control over the effective number of decoupled sectors as a function of time $N_{\text{sectors}}(t)$. We consider both situations of a perturbation that leads to 'local' and 'non-local' exploration of the sectors. We then turn to the complex SYK model with an approximate $U(1)$ symmetry broken by a charge $q_0 = 2$ operator with small coefficient in the Hamiltonian.

## 3.1 Local exploration of Hilbert space

Following (2.5) we construct a block diagonal matrix $H_0$ with $Q$ blocks of size $N_q \times N_q$ sampled from GUE, corresponding to the conserved charge sectors. The symmetry breaking perturbation $V$ is a matrix connecting only sectors of charge $q$ and $q \pm 1$, with periodic boundary conditions such that $Q + \alpha \equiv \alpha$. The hermitian and anti-hermitian parts of the blocks $V_q$, $q = 1, 2, \ldots, Q$ (see (2.5)) are sampled from GUE. We then study numerically the spectral form factor $\text{SFF}(t)$ for $H_0 + \epsilon V$ and its relaxation towards the late time expectation $\text{SFF}_{\text{RMT}}(t)$ for a random $QN_q \times QN_q$ matrix.

The results are shown in Fig. 2 for different values of $\epsilon$. We find excellent agreement between numerics and the analytic prediction from Hilbert space diffusion, Eqs. (2.4) with $N_{\text{sectors}}(t)$ given by (2.17) and diffusivity $\Gamma$ given by (2.13). Note that this agreement is obtained without any fitting parameter. Using the full analytic prediction (2.17) rather than its continuum approximation (2.19) accounts for the smooth behaviour at the onset and end of diffusion.

We briefly comment on the expected regime of validity of our predictions. Clearly, $\epsilon$ needs to be small enough for weak symmetry-breaking to be the dominant bottleneck towards

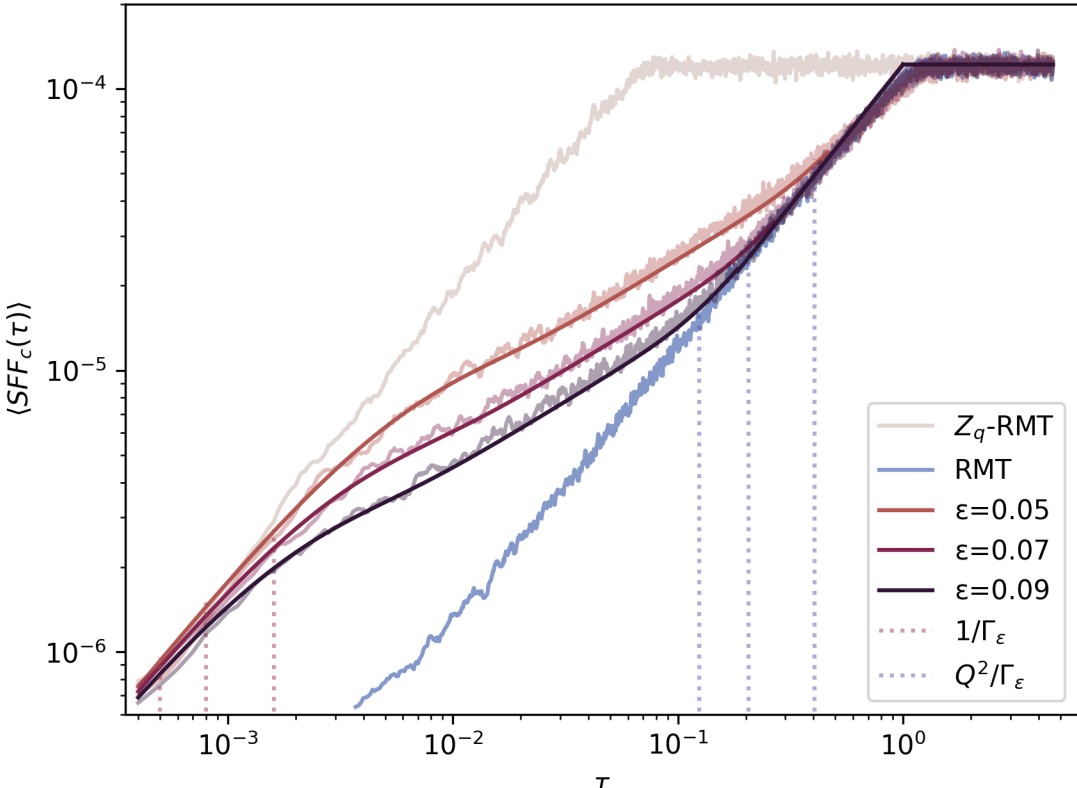

Figure 2: Spectral form factor for the RMT model with approximate $\mathbb{Z}_Q$ symmetry, showing the smooth transition from $Q = 16$ GUE charge sectors to full $(QN_q)$-dimensional RMT with $N_q = 512$ via local exploration of Hilbert space. The transition is governed by the effective number of decoupled sectors which follows an approximate $1/\sqrt{t}$ behaviour, (2.18). The light noisy curves correspond to the numerically obtained SFFs, averaged over 100 realizations. The dark smooth curves come from the analytic prediction (2.17) with diffusivity given by (2.13), for three different values of $\epsilon$. The dashed red and blue lines denote the onset and end of Hilbert space diffusion, respectively.

establishing the RMT regime. Conversely, if $\epsilon$ is too small to affect level spacing, it will not have appreciable effects on the SFF. These two conditions read

$$t_{\text{Th},0} \ll \frac{1}{4\pi\Gamma} \ll t_{\text{Heis},0} . \tag{3.1}$$

In the case of RMT, the Thouless time is replaced by the microscopic timescale set by the width of the spectrum $t_{\text{Th},0} \to 1/\Delta E$. Substituting $\Gamma$ with Eq. (2.13), using $t_{\text{Heis}, H_0} = 2\pi N_q/\Delta E$, and with the normalization $|V|^2 = 1/N_q$, the regime of validity of our results is $1/\sqrt{N_q} \ll \frac{\sqrt{2}2\pi}{\Delta E}\epsilon \ll 1$. For the parameters used in Fig. 2, $N_q = 512$, $\Delta E = 4$, this becomes

$0.0079 \ll \epsilon \ll 0.45$.

## 3.2 Non-local exploration of Hilbert space

One can also consider an RMT model with non-local exploration of approximate symmetry sectors. The perturbation $V$ is in this case a full $N_q \times N_q$ matrix sampled from GUE, connecting any two sectors $q, q' \in \{1, 2, \ldots, Q\}$. The results are shown in Fig. 3, showing again excellent agreement between numerics and theory, without any fitting parameter. The effective number of sectors entering in the theory prediction $\text{SFF}(t) = N_{\text{sectors}}(t)\text{SFF}_{\text{RMT}}(t)$ is now given by (2.21), with rate $\Gamma$ still given by (2.13).

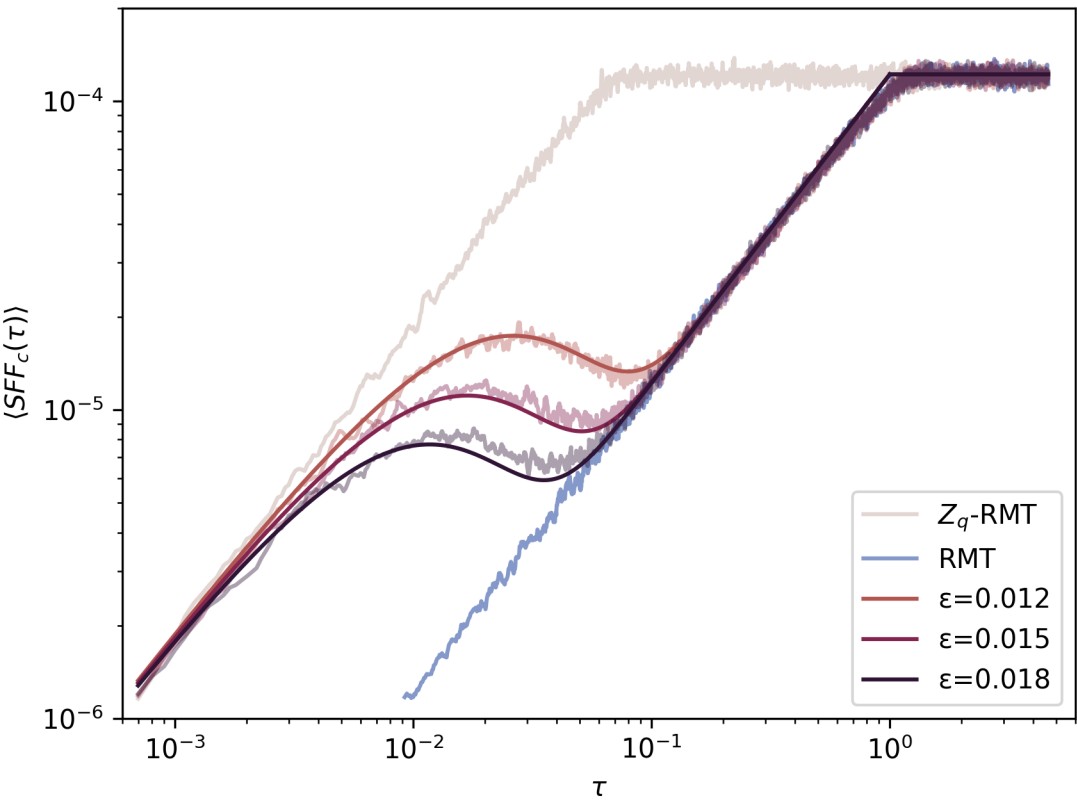

Figure 3: Spectral form factor for the RMT model with approximate $\mathbb{Z}_Q$ symmetry, where the symmetry-breaking perturbation $\epsilon V$ connects any two sectors, leading to faster 'non-local' exploration of the Hilbert space. Due to the faster approach to full RMT, smaller values of $\epsilon$ are shown. Parameters are otherwise identical to those used in Fig. 2.

As in the previous section, we can estimate the regime of validity of our description. The

onset of exploration of the full Hilbert space now occurs around the timescale $1/Q\Gamma$, so that the condition (3.1) becomes,

$$t_{\text{Th},0} \ll \frac{1}{Q\Gamma} \ll t_{\text{Heis},0}\,, \qquad (3.2)$$

which for the parameters used in Fig. 3 becomes $0.007 \ll \epsilon \ll 0.40$.

## 3.3   Complex SYK with approximate $U(1)$ symmetry

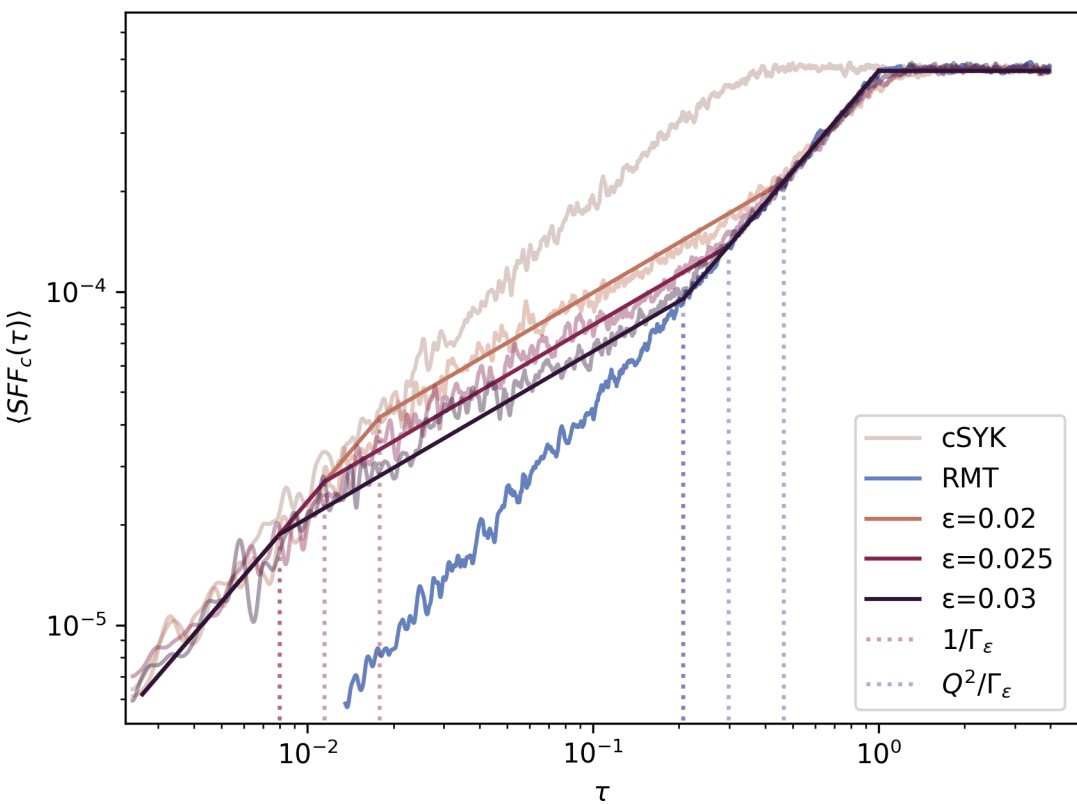

Figure 4: Numerically computed connected SFF for complex SYK with $Q = 13$ averaged over 100 realisations. The SFF presented in the plot is computed by summing over all even numbered parity sectors. The solid lines show the analytically predicted $\sqrt{t}$-behaviour via local diffusion for intermediate timescales. The solid lines were fitted using the $\Gamma$ prediction from Eq. (2.13) and the number of sectors from Eq. (2.18). We find a good agreement with the theoretical prediction of Sec. 2 up to a constant $\mathcal{O}(1)$ factor. Since the mean level spacing varies significantly between the centre of the spectrum and its edge, we have used a Gaussian filter to suppress the contribution of the eigenvalues near the edge of the spectrum.

Having confirmed our predictions in two toy RMT models, we now turn to a physical many-

body system with an approximate symmetry: the complex SYK model, with weak explicit breaking of the $U(1)$ symmetry. We consider the following Hamiltonian:

$$H = H_0 + \epsilon V = \sum_{\substack{i<j \\ k<l}} J_{ijkl} c_i^\dagger c_j^\dagger c_k c_l + \epsilon \sum_{\substack{i \\ j<k<l}} \left( \widetilde{J}_{ijkl} c_i^\dagger c_j c_k c_l + \text{h.c.} \right) , \qquad (3.3)$$

with complex fermions satisfying $\{c_i^\dagger, c_j\} = \delta_{ij}$ for $i,j = 1,2,\ldots,Q$. The unperturbed Hamiltonian ($\epsilon = 0$) describes the complex SYK model (see, e.g., Ref. [36]), with couplings chosen randomly from a Gaussian distribution with zero mean and $\overline{|J_{ijkl}^2|} = \overline{|\widetilde{J}_{ijkl}^2|} = \frac{3! J^2}{N^3}$, where $J$ sets the average strength of the coupling. This unperturbed model has a $U(1)$ symmetry $c_j \to e^{i\theta} c_j$, which separates the Hilbert space into $Q+1$ sectors of $U(1)$ charge $q = 0,1,\ldots,Q$, and size $N_q = \binom{Q}{q}$. The total Hilbert space dimension is $\sum_q N_q = 2^Q$.

Adding the charge $q_0 = 2$ operator to the Hamiltonian with small non-zero coefficient $\epsilon \ll 1$ explicitly breaks the $U(1)$ symmetry down to a $\mathbb{Z}_2$ symmetry measuring fermion parity. Because the symmetry-breaking deformation only connects nearby charge sectors $q \to q \pm 2$, we expect the SFF of this model to feature diffusive growth $\sim \sqrt{t}$ at intermediate times, characteristic of 'local' exploration of the Hilbert space as discussed in Sec. 2. Unlike in the previous examples, the size of each block $N_q$ now depends on $q$, so that strictly one should consider diffusion on an inhomogeneous lattice, with site-dependent diffusivity. This is simple to incorporate (and is discussed in Sec. 4.3.2). However, the $q$-dependence of $N_q$ can approximately be ignored at large $Q$, where observables are dominated by the largest sectors near half-filling $q \sim \frac{Q}{2} \pm \sqrt{Q}$, with width set by the width of the binomial distribution. We therefore still expect the SFF to feature the $\sqrt{t}$ behaviour, with a reduced number of sectors $Q_{\text{eff}} \sim \sqrt{Q}$. Fig. 4 shows that the numerically obtained SFF indeed meets this expectation. However, the effect of different block sizes presents itself in the SFF of the un-deformed model as the smoothening of signal when the ramp transitions into plateau. This is a consequence of different Heisenberg time corresponding to each block of different size.

# 4 Hilbert space diffusion: a $\sigma$-model approach

The ergodic behaviour of quantum chaotic systems can be described using an effective field theory (EFT) of light degrees of freedom governed by a $\sigma$-model on a (super-)coset manifold [37–39] (see Refs. [33, 40] for reviews). In this section, our goal is to extend these EFTs to capture the diffusive behaviour of theories with approximate symmetries uncovered in

section 2. Specifically, we will again consider a Hamiltonian of the form (2.5),

$$H = H_0 + \epsilon V \ , \tag{4.1}$$

where, $H_0$ is the unperturbed Hamiltonian with $\mathbb{Z}_Q$ symmetry and $V$ is the perturbation that breaks this symmetry. The structure of $V$ is such that it connects sectors of charge $q$ with adjoining sectors of charge $q \pm 1$, leading to 'local' exploration of the Hilbert space. Following the discussion in section 2, we expect this perturbation to allow for gradual diffusive-like exploration of the Hilbert space, leading to a characteristic $\sqrt{t}$ behaviour of the SFF at intermediate times (Fig. 1).

We start by briefly reviewing the framework of the chaotic sigma model, before adapting the formalism in order to incorporate the local Hilbert-space diffusion we just reviewed.

## 4.1 General framework: chaotic $\sigma$-models

When one studies a system with a known Hamiltonian, one of the first things to do is to solve for its energy spectrum. Of course, most of the time it is not possible to solve the eigen-equation exactly, but nevertheless it still makes sense to talk about a (-n energy) density of states $\rho(E)$. When studying the spectral properties of a system, a useful observable describing these properties is given by the spectral resolvent, $R_{\pm}(z)$.[5] Defined in terms of the retarded/advanced Green's function, it is written as

$$R_{\pm}(z) = \text{Tr}[G_{\pm}(z)] = \text{Tr}[z \pm i\delta - H]^{-1}. \tag{4.2}$$

The resolvent is related to the spectral density via the formula $\rho(z) = \mp\frac{1}{\pi}\text{Im}\, R_{\pm}(z)$. The resolvent can also be obtained from the generating functional,

$$\mathcal{Z}_{(2)}(z_1, z_2) = \frac{\det(z_2 - H)}{\det(z_1 - H)} \ , \tag{4.3}$$

by differentiating with respect to the energy argument $z_1$, and then setting $z_1 = z_2$,

$$\rho(E) = \mp\frac{1}{\pi}\text{Im}\, \partial_{z_1} \mathcal{Z}_{(2)}(z_1, z_2)\Big|_{z_1 = z_2 = E \pm i0}. \tag{4.4}$$

---

[5]We shall use the notation $z^{\pm}$ for denoting a small imaginary, positive/negative energy offset $z^{\pm} = z \pm i\delta$.

We notice also the symmetry between the energy arguments, $z_i$. Differentiating instead with respect to $z_2$ will change the result only by a minus sign.

The two-point function of the density of states can likewise be computed using the following generating function,[6]

$$\left\langle \rho(E)\rho(E')\right\rangle_c = \frac{1}{2\pi^2}\text{Re}\,\partial_{z_1}\partial_{z_2}\left\langle \mathcal{Z}_{(4)}(z_3, z_4, z_1^+, z_2^-)\right\rangle_c \bigg|_{\substack{z_3=z_1^+=E \\ z_4=z_2^-=E'}}, \tag{4.5}$$

where,

$$\mathcal{Z}_{(4)} = \frac{\det(z_3 - H)\det(z_4 - H)}{\det(z_1^+ - H)\det(z_2^- - H)}. \tag{4.6}$$

Note that in the above equation we have written the product of Green's functions on the RHS inside angle brackets, $\langle \cdot \rangle$.[7] These angle brackets represent coarse-graining/averaging that can be taken over different quantities, for example over statistical ensembles or over microcanonical energy windows.[8] The averaging/coarse-graining is not a necessity, but useful to extract the self-averaging part of the physical observables in any theory. The choice of the imaginary offset of the energy arguments in the denominator gives us the required product of density of states. From the connected part of the two-point function $R_2(E,\omega) = \left\langle \rho(E + \frac{\omega}{2})\rho(E - \frac{\omega}{2})\right\rangle_c$, the connected spectral form factor is obtained by taking a Fourier transform,

$$\text{SFF}(t) = \frac{1}{\rho(E)^2}\int d\omega\, R_2(\omega)e^{-i\omega t}. \tag{4.7}$$

The definition of SFF in (4.7) might appear to be different from the one in section 2. Here the SFF has an explicit dependence on the energy $E$. However, when defined within a microcanonical ensemble around energy $E$, the two definitions agree. If one is interested in computing the SFF over the entire spectrum, then one can integrate the expression defined in (4.7).

Having set up our framework, we now explain how to compute the quantities introduced above explicitly using the supersymmetric approach of Refs. [37–39]. Our main goal is to compute $\mathcal{Z}_4$. The inverse determinants can be rewritten as integrals over bosonic variables $S_i$, and a similar rewriting is possible for the determinants in the numerator where we use

---

[6]The principle part of the Green's functions contributes only to the disconnected part and would subsequently be dropped.

[7]The subscript $c$ denotes the corrected part of the correlation function that is defined as $\left\langle AB\right\rangle_c = \left\langle AB\right\rangle - \left\langle A\right\rangle\left\langle B\right\rangle$

[8]In SYK for example, one averages over different disordered realizations.

fermionic valued field variables, $\chi_i$. For a simpler representation of these integrals, we regroup both field types into a single $4L$-dimensional graded vector,

$$\psi = (S_i^r, S_i^a, \chi_i^r, \chi_i^a)^T \quad \text{and} \quad \bar{\psi} = (\bar{S}_i^r, \bar{S}_i^a, \bar{\chi}_i^r, \bar{\chi}_i^a) \; . \tag{4.8}$$

Here the bar represents a generalised adjoint operation, $\bar{\psi} = \psi^\dagger \cdot g$, on the graded vector.[9] The matrix, $g = \text{diag}(1, -1, 1, 1)$, is required for the convergence of the integral (4.10) defined below. The indices $r/a$ stand for retarded/advanced and the lower index $i$ runs over the Hilbert space, $i = 1, \ldots, L$. In general, the total vector space can be split into a tensor product of three subspaces, representing the graded, retarded/advanced, and the physical Hilbert space nature,

$$\mathcal{H} = \underbrace{\mathcal{H}^{bf} \otimes \mathcal{H}^{ra}}_{4-\text{dim.}} \otimes \underbrace{\mathcal{H}^i}_{L-\text{dim.}} \; . \tag{4.9}$$

For notational purposes, we'll only write the indices over which we currently sum over; or when it's not clear from the context to which we are referring. We can now set up the partition function as Gaussian integral,

$$\mathcal{Z}_{(4)}[\hat{z}] = \int d(\bar{\psi}, \psi) e^{i\bar{\psi}(\hat{z} - \hat{H})\psi}. \tag{4.10}$$

We use the $\hat{\cdot}$ notation to denote matrices in the full $4L$-dimensional Hilbert space. Thus, $\hat{H}$ is equivalent to one copy of the Hamiltonian $H$ in each graded sector such that $\hat{H} = \mathbb{1}^{bf} \otimes \mathbb{1}^{ra} \otimes H$. Moreover, we have regrouped the four energy variables $z_i$ into a diagonal graded four by four matrix denoted by $\hat{z}$. The role of the energy variables $z_i$ is multifold. As one sees in equation (4.5), the $z_i$ source the insertions of the Green's functions. Additionally, careful choice of the imaginary parts ensures convergence of the Gaussian integrals. At the same time, it governs the spontaneous causal symmetry breaking of the action in equation (4.10).[10] Once the sources are turned off, $\hat{z}$ takes the following form in the action,

$$\hat{z} = z \otimes \mathbb{1}^i = \mathbb{1}^{bf} \otimes \left( E\mathbb{1}^{ra} + \left( \tfrac{\omega}{2} + i\delta \right) \sigma_3^{ra} \right) \otimes \mathbb{1}^i \; . \tag{4.11}$$

For notational simplicity we will not always write out the full Hilbert space nature, but it will be clear from context. Also, in this notation, it is even clearer that this machinery can

---

[9]The generalisation of adjoint fields permits us to generalise the following analysis to the different symmetry groups from the 10-fold symmetry classification of RMT [41–43].

[10]This was referred to as 'causal symmetry breaking' in [6], a terminology we will also adopt in this work. The nomenclature makes reference to the fact that the different signs of the $i\epsilon$ prescription are related to the retarded and advanced causal Green's functions, respectively.

be easily generalized to higher point correlation functions.

Let us now discuss the symmetry structure of this action. In our case, the fields $(\bar{\psi}, \psi)$ transform under a general transformation $T \in U(L, L|2L)$. The presence of $\hat{H}$ breaks this symmetry explicitly to $U(1, 1|2) \times \mathbb{1}_L \equiv U(1, 1|2)$. This is the reason the EFT is defined in terms of a four by four graded matrices, denoted later by $A$; and is independent of the dimension, $L$, of our theory. Furthermore, the identity contribution proportional to $E$ in $\hat{z}$, (4.11), is irrelevant for any further symmetry breaking, but the second part $\left(\frac{\omega}{2} + i\delta\right) \sigma_3^{ra}$ acting as Pauli matrix in the retarded and advanced space determines the weak and spontaneous breaking of the causal symmetry to the group: $U(1|1) \times U(1|1)$. According to this symmetry breaking pattern, the degrees of freedom important for describing the EFT are the (pseudo-) Goldstone modes that live in the manifold $U(1, 1|2)/(U(1|1) \times U(1|1))$.

## 4.2 Effective field theory of Hilbert space diffusion

As we alluded to above, to extract the self-averaging part of the SFF that is described by the symmetry-broken theory of the chaotic sigma model, one is required to average the generating function over an appropriate ensemble. In our toy-model, we average over the Hamiltonians, with a measure $\int P(H)(\cdot)dH$. This average splits in our case into two independent averages over the diagonal block matrices $H_0$ and over the perturbation matrices $V$. We remind the reader that we constructed the entries of both of these matrices to be sampled from a GUE ensemble. We define therefore the distributions,

$$P(H_0) = \prod_\alpha P(H_0^{\alpha\alpha}) = \prod_\alpha e^{-\frac{N_q}{2\sigma}\operatorname{Tr}[H_0^{\alpha\alpha}(H_0^{\alpha\alpha})^\dagger]} \ , \tag{4.12}$$

$$P(V) = \prod_{\alpha<\beta} P(V^{\alpha\beta}) = \prod_{\alpha<\beta} e^{-\frac{N_q}{\sigma}\operatorname{Tr}[V^{\alpha\beta}(V^{\alpha\beta})^\dagger]} \ . \tag{4.13}$$

In our case, since we're restricting to nearest neighbour interactions, $\beta = \alpha + 1$ and the index $\alpha$ runs over the number of charges, $\alpha = 1, \cdots, Q$. We are imposing periodic boundary conditions, such that $\alpha + Q \equiv \alpha$. The size of an individual charge sector is denoted by $N_q$. Hence, the dimensionality of the total Hilbert space, $L = QN_q$. Now we have all the ingredients to compute the generating function, given by the formula,

$$\left\langle \mathcal{Z}_{(4)}[\hat{z}] \right\rangle_H = \int d(\bar{\psi}, \psi)dH P(H) e^{i\bar{\psi}(\hat{z}-\hat{H})\psi}. \tag{4.14}$$

### 4.2.1 $\sigma$-model action

After expanding the action (4.14) in the fields, restricting only to the non-zero terms, $\alpha, \beta = \alpha + 1$ and averaging over GUE sampled Hamiltonians $H$ given in (4.12) we find,

$$\langle \mathcal{Z}_{(4)}[z] \rangle = \int d(\bar{\psi}, \psi) e^{-\frac{\sigma}{2N_q} \sum_\alpha \mathrm{Str}\left[(M^\alpha)^2 + 2\epsilon^2 M^\alpha M^{\alpha+1}\right]} e^{i \sum_\alpha \mathrm{Str}[z^\alpha M^\alpha]}. \tag{4.15}$$

We have introduced the supertrace notation with $M^\alpha$ a four by four Hermitian supermatrix,[11] with two diagonal blocks, one accounting for the retarded, one for the advanced sector,

$$M^\alpha = \begin{pmatrix} M_r^\alpha & 0 \\ 0 & M_a^\alpha \end{pmatrix} \quad \text{where} \quad M_I^\alpha = \begin{pmatrix} \bar{S}_I^\alpha S_I^\alpha & \bar{\chi}_I^\alpha S_I^\alpha \\ \bar{S}_I^\alpha \chi_I^\alpha & \bar{\chi}_I^\alpha \chi_I^\alpha \end{pmatrix}, \quad I = a, r. \tag{4.16}$$

In $M_I^\alpha$ the Hilbert space indices $i$ are summed over. Now, before proceeding, let's analyse the action a bit more in detail. The first term $\mathrm{Str}\left[(M^\alpha)^2 + 2\epsilon^2 M^\alpha M^{\alpha+1}\right]$ is the action of a one-dimensional lattice model with nearest neighbour interactions, where at each site $\alpha$ we have a supermatrix valued field. The second piece, $\mathrm{Str}\left[z^\alpha M^\alpha\right]$, can be thought of as an external magnetization, which we have generalised to a location-dependent field, $z^\alpha$. Following this idea, we rewrite the equation (4.15) in a more compact way with the use of a $Q \times Q$ matrix $K(\epsilon^2)$ with periodic boundary conditions

$$K(\epsilon^2) = \begin{pmatrix} 1 & \epsilon^2 & 0 & \cdots & & \epsilon^2 \\ \epsilon^2 & 1 & \epsilon^2 & & & \\ 0 & \epsilon^2 & 1 & & & \\ \vdots & & & \ddots & & \epsilon^2 \\ \epsilon^2 & & & & \epsilon^2 & 1 \end{pmatrix}, \tag{4.17}$$

so that

$$\langle \mathcal{Z}_{(4)}[z] \rangle = \int d(\bar{\psi}, \psi) e^{i\mathrm{Str}[Z^\mathrm{T} M]} e^{-\frac{\sigma}{2N_q} \mathrm{Str}\left[M^\mathrm{T} K(\epsilon^2) M\right]}. \tag{4.18}$$

Here the $M^\alpha$ matrices are ordered as entries in a vector $M = (M^1, \ldots M^Q)^\mathrm{T}$, as well as the energies $z^\alpha$ as entries in $Z = (z^1, \ldots z^Q)^\mathrm{T}$. The identity elements on the diagonal of $K(\epsilon)$ produce the mass term, $(M^\alpha)^2$, and the off-diagonal elements giving the nearest neighbour interactions.

Next, we perform the integral over the superfields $\bar{\psi}, \psi$. To do so, we use the Hubbard-

---

[11]The Hermititian conjugate of supermatrices is defined differently from that for standard matrices, $M^\dagger = \begin{pmatrix} a & \xi \\ \zeta & b \end{pmatrix}^\dagger = \begin{pmatrix} a^* & \zeta^* \\ -\xi^* & b^* \end{pmatrix}$. Along with the property of complex conjugation for Grassmann numbers, $(\xi^*)^* = -\xi$, the Hermiticity of $M^\alpha$ follows.

Stratonovich (HS) trick to decouple the quartic interactions in the $(M^\alpha)^2$ term,

$$e^{-\frac{\sigma}{2N_q}\mathrm{Str}[M^{\mathrm{T}}K(\epsilon^2)M]} = \int dA\, e^{-\frac{N_q}{2\sigma}\mathrm{Str}[A^{\mathrm{T}}K(-\epsilon^2)A]+i\mathrm{Str}[A^{\mathrm{T}}M]} . \tag{4.19}$$

Here, we have introduced an HS field at each site $\alpha$, which collectively written as $A = (A^1, \ldots A^Q)^{\mathrm{T}}$. Subsequently, performing the integration over the $(\bar\psi, \psi)$ fields, which is a Gaussian supermatrix integration, leads to an integral over the four-by-four dimensional matrices $A$,

$$\langle \mathcal{Z}_{(4)}[z]\rangle = \int dA\, e^{-\frac{N_q}{2\sigma}\sum_\alpha \mathrm{Str}\left[(A^\alpha)^2 - 2\epsilon^2 A^\alpha A^{\alpha+1}\right] - N_q \sum_\alpha \mathrm{Str}\ln[z^\alpha + A^\alpha]} \equiv \int dA\, e^{-N_q S[A]} . \tag{4.20}$$

The supertraces are also only over the four-by-four graded space. Since each graded sector contains $N_q$-fields, an overall $N_q$ factor appears also in front of the ln-term. This means the whole exponent has an overall $N_q$ factor, which allows us to solve the integral via saddle-point analysis for $N_q \to \infty$. Note that the HS fields $A^\alpha$ transform under the adjoint representation of $U(1,1|2)$, which follows from the transformation properties of $\bar\psi, \psi$ fields.

### 4.2.2 Stationary phase analysis

In the large $N_q$ limit, the saddle point equations corresponding to $A^\alpha$ are given by,

$$\frac{1}{z^\alpha + A^\alpha} + \frac{1}{\sigma}\left[A^\alpha - \epsilon^2 A^{\alpha+1} - \epsilon^2 A^{\alpha-1}\right] = 0 . \tag{4.21}$$

Solving this matrix equation for general $\epsilon, \omega$ values is difficult, we therefore opt for a perturbative study. For the $\epsilon = \omega = 0$, we find (recall that $z = E\mathbb{1}^{bf} \otimes \mathbb{1}^{ra}$ when $\omega = 0$, see Eq. (4.11))

$$A_0^\alpha = -\tfrac{1}{2}E\mathbb{1}^{bf} \otimes \mathbb{1}^{ra} + i\gamma(E)\,\tilde\Lambda_g^\alpha \quad \text{with} \quad \gamma(E) = \sqrt{\sigma - \tfrac{E^2}{4}} , \tag{4.22}$$

and where $\tilde\Lambda^\alpha$ is a hermitian matrix that is a square root of the identity $\mathbb{1}^{bf} \otimes \mathbb{1}^{ra}$. In the absence of the interaction, $\epsilon = 0$, we have an independent saddle point equation for each $\alpha$. The matrix $\tilde\Lambda$ can be found from the orbits of base solutions $\tilde\Lambda_0$ under the pseudo-unitary group, $\tilde\Lambda_g = g\tilde\Lambda_0 g^{-1}$, with $g \in U(1,1|2)$. Choosing $g$ to diagonalize $\tilde\Lambda$, it is sufficient to consider as base solutions $\tilde\Lambda_0$ diagonal matrices that are square roots of unity: $\tilde\Lambda_0 = \mathrm{diag}(\pm 1, \pm 1, \pm 1, \pm 1)$. This leads to 16 different solutions; however, not all solutions contribute, since only a few can be reached by deforming the initial integration contour of

Eq. (4.20). If we concentrate first on the bosonic sector, we have to pay attention to the poles in the complex $A^\alpha$ plane, stemming from the term $e^{-\text{Strln}[z^\alpha + A^\alpha]} = \text{Sdet}[z^\alpha + A^\alpha]^{-1}$. The original integration contour along the real axis can be deformed to reach either the saddle in the upper or lower half plane in order to avoid the poles, but not both at the same time. On the other hand, in the fermionic sector there are no poles and the initial integration contour along the imaginary axis can be deformed such that it passes through both saddles and the zero. The saddle point corresponding to the choice,

$$\tilde{\Lambda}_0 = \mathbb{1}^{bf} \otimes \sigma_3^{ra} \equiv \Lambda \,, \tag{4.23}$$

has the non-perturbatively leading contribution when $\omega \neq 0$ and is conventionally referred to as the standard saddle point. Since we are interested in $\omega \neq 0$, we begin with standard saddle point solution for all charge sectors, $\alpha$,

$$A_0^\alpha = -\tfrac{1}{2} E \mathbb{1}^{bf} \otimes \mathbb{1}^{ra} + i\gamma(E)\,\Lambda, \quad \forall \alpha \,. \tag{4.24}$$

The second saddle point corresponding to,

$$\tilde{\Lambda}_0 = \sigma_3^{bf} \otimes \sigma_3^{ra} \equiv \Lambda_{\text{AA}} \,, \tag{4.25}$$

is referred to as Andreev-Altshuler (AA) saddle point [12]. It lies in the pseudo-unitary group orbit of the standard saddle point. The remaining two saddle points are truly subleading in the large $L$ limit [44]. For a detailed analysis on how to choose this dominant saddle point solution, we refer to [23, 33, 40].

Now we have all the ingredients to talk about the symmetry breaking procedure. Going back to the action in (4.20), we can see that for $\omega = 0$ and $\epsilon = 0$, the action is invariant under a symmetry transformation $T \in U(1, 1|2)^Q$. But by choosing a specific saddle, as (4.23) does, the symmetry of the full action gets spontaneously broken down to $\left(U(1|1) \times U(1|1)\right)^Q$. And the resulting coset manifold is therefore,

$$\underset{\alpha}{\otimes} \mathcal{M}^\alpha = \left( \frac{U(1, 1|2)}{U(1|1) \times U(1|1)} \right)^Q \,. \tag{4.26}$$

This symmetry is explicitly broken when $\omega \neq 0$. When the global symmetry of the original theory is broken, $\epsilon \neq 0$, the fields, $A^\alpha$, corresponding to individual sectors start interacting. In this case, the symmetry is further broken down to $U(1|1) \times U(1|1)$. The "pseudo-

Goldstone" modes that belong to the coset manifold,

$$\mathcal{M}_q = \frac{\mathrm{U}(1,1|2)^Q}{\mathrm{U}(1|1) \times \mathrm{U}(1|1)} \ , \tag{4.27}$$

are the lightest modes governing the universal physics of our EFT. These modes control the fluctuations around the dominant saddle point solution and are parameterised by the elements of the coset manifold $U^\alpha \in \mathcal{M}^\alpha$,

$$A^\alpha = -\tfrac{1}{2}E\mathbb{1} + i\gamma(E)\, U^\alpha \Lambda^\alpha \left(U^\alpha\right)^{-1} \ . \tag{4.28}$$

Before we proceed, we would like to mention that so far we had been working with the 'Bose-Fermi' convention in which the graded space is decomposed as $(bf) \otimes (ra)$. However, for parametrising the supercoset manifold, it is more convenient to decompose in the 'advanced-retarded' convention, $(ra) \otimes (bf)$, which we use henceforth.[12] The above fluctuations can be further written in an exponential representation in terms of "pion" fields $B^\alpha, \tilde{B}^\alpha$,

$$U^\alpha = e^{-W^\alpha}, \quad W^\alpha = -\left(\begin{smallmatrix} 0 & B^\alpha \\ \tilde{B}^\alpha & 0 \end{smallmatrix}\right) \ , \tag{4.29}$$

and expanded in a weak-field approximation,

$$U^\alpha \Lambda^\alpha (U^\alpha)^{-1} = \Lambda^\alpha + 2\Lambda^\alpha W^\alpha + 2\Lambda^\alpha (W^\alpha)^2 + \dots \ . \tag{4.30}$$

The one-loop exactness of the $\sigma$-model action on $\mathcal{M}_q$ implies that we need to consider only up to quadratic terms in $B^\alpha, \tilde{B}^\alpha$.[13] Carrying out this expansion of the full action (4.20) leads us to,

$$
\begin{aligned}
S[A] &= \sum_\alpha S_\alpha \,, \\
S_\alpha &= \mathrm{Strln}[E\mathbb{1} + A_0^\alpha] + \tfrac{E}{2\sigma}\mathrm{Str}[\left(\begin{smallmatrix} \omega_1 & 0 \\ 0 & \omega_3 \end{smallmatrix}\right) - \left(\begin{smallmatrix} \omega_2 & 0 \\ 0 & \omega_4 \end{smallmatrix}\right)] - \tfrac{i\gamma}{\sigma}\mathrm{Str}[\left(\begin{smallmatrix} \omega_1 & 0 \\ 0 & \omega_3 \end{smallmatrix}\right) + \left(\begin{smallmatrix} \omega_2 & 0 \\ 0 & \omega_4 \end{smallmatrix}\right)] \\
&\quad - \tfrac{2i\gamma}{\sigma}\mathrm{Str}[B^\alpha \tilde{B}^\alpha \left(\begin{smallmatrix} \omega_1 & 0 \\ 0 & \omega_3 \end{smallmatrix}\right) - \tilde{B}^\alpha B^\alpha \left(\begin{smallmatrix} \omega_2 & 0 \\ 0 & \omega_4 \end{smallmatrix}\right)] + \tfrac{4\epsilon^2\gamma^2}{\sigma}\mathrm{Str}[(B^{\alpha+1} - B^\alpha)(\tilde{B}^{\alpha+1} - \tilde{B}^\alpha)] + \dots \ .
\end{aligned}
\tag{4.31}
$$

In the above equation, $\cdots$ represents all higher order terms in $B^\alpha, \tilde{B}^\alpha$. We refer to the appendix [A] for details of the computation. We represent by $S[B, \tilde{B}]$ the part of the action

---

[12]This corresponds to permuting the second and the third rows and columns in the supermatrices.

[13]Normally, one would need to include the contributions of one-loop path integral around both the saddle points. However, the AA saddle point gives rise to the plateau *after* the Heisenberg time and the contribution of the standard saddle is sufficient to reproduce the ramp [33, 40]. The Hilbert space diffusion occurs at time scales earlier than $t_H$ so we don't consider the contribution of the AA saddle point for current purposes.

generating the diffusion equation for the $B$-fields,

$$S[B, \tilde{B}] = -\frac{2i\gamma}{\sigma}\text{Str}[B^\alpha \tilde{B}^\alpha \left(\begin{smallmatrix} \omega_1 & 0 \\ 0 & \omega_3 \end{smallmatrix}\right) - \tilde{B}^\alpha B^\alpha \left(\begin{smallmatrix} \omega_2 & 0 \\ 0 & \omega_4 \end{smallmatrix}\right)] + \frac{4\epsilon^2\gamma^2}{\sigma}\text{Str}[(B^{\alpha+1} - B^\alpha)(\tilde{B}^{\alpha+1} - \tilde{B}^\alpha)]. \quad (4.32)$$

For our perturbative analysis it is sufficient to consider only lowest order terms in $\omega$ and $\epsilon$.

Recall that the energy arguments, $\omega_1$ and $\omega_2$, source the insertion of retarded and advanced Green's functions, (4.2), respectively. On differentiating with respect to these sources, we obtain an insertion in terms of the pions in the above integral,

$$\left\langle G_+\left(E + \frac{\omega}{2}\right) G_-\left(E - \frac{\omega}{2}\right)\right\rangle = 4\left(\frac{N_q\gamma}{\sigma}\right)^2 \left\langle \sum_{\alpha,\alpha'} \text{Str}\left[B_\alpha \tilde{B}_\alpha P_b\right]\text{Str}\left[\tilde{B}_{\alpha'} B_{\alpha'} P_b\right]\right\rangle + \frac{N_q^2}{\sigma} .$$

$$(4.33)$$

Here, $P_b$ is the projection operator on the bosonic sector. The second term is the contribution of the disconnected part, and so we drop it subsequently. After having taken the derivative with respect to the sources, we obtain the effective action of our EFT by setting $\omega_3 = \omega_1$ and $\omega_4 = \omega_2$,

$$S[B, \tilde{B}] = -\frac{2\gamma}{\sigma}\sum_{\alpha=1}^{Q}\left(i(\omega_1 - \omega_2)\text{Str}[B^\alpha \tilde{B}^\alpha] - 2\epsilon^2\gamma\text{Str}[(B^{\alpha+1} - B^\alpha)(\tilde{B}^{\alpha+1} - \tilde{B}^\alpha)]\right) \quad (4.34)$$

We rewrite the above action in the momentum space corresponding to the discrete $\alpha$-lattice,

$$S[b, \tilde{b}] = -\frac{2\gamma}{\sigma}\sum_{n=0}^{Q-1}\left(i(\omega_1 - \omega_2) - \Gamma_n\right)\text{Str}\left[b_n \tilde{b}_n\right] , \quad (4.35)$$

$$\text{where, } B_\alpha = \frac{1}{\sqrt{Q}}\sum_{n=0}^{Q-1}e^{2\pi i\alpha\frac{n}{Q}}b_n, \qquad \tilde{B}_\alpha = \frac{1}{\sqrt{Q}}\sum_{n=0}^{Q-1}e^{-2\pi i\alpha\frac{n}{Q}}\tilde{b}_n, \quad (4.36)$$

and $\Gamma_n = 8\epsilon^2\gamma\sin^2\left(\frac{\pi n}{Q}\right)$, which as we will see shortly matches the expression found in Eq. (2.16).

## 4.3 Hilbert space diffusion in $\sigma$-model

The equation of motion for $B_\alpha$ can be found by varying the action $\delta S[B]/\delta \tilde{B}_\alpha = 0$ (the $\tilde{B}_\alpha$ equation is similar):

$$-i(\omega_1 - \omega_2)B_\alpha = 2\epsilon^2 \gamma (B_{\alpha+1} - 2B_\alpha + B_{\alpha-1}). \tag{4.37}$$

The ensuing physics is transparent: we can directly read off a discrete derivative in $\alpha$-space, $B_{\alpha+1} - 2B_\alpha + B_{\alpha-1} = d_\alpha^2 B_\alpha$. Also, by Fourier transforming the above equation to the time domain, $-i(\omega_1 - \omega_2)$ makes a continuous time derivative $\partial_t$ apparent, and we find a diffusion equation for the fields $B_\alpha \to B(\alpha, t)$, in accordance with our earlier phenomenological discussion. Moreover, by keeping track of the factors in front of $\omega_i$ and $B_\alpha$ we determine the diffusion constant $\Gamma$, such that,

$$\partial_t B(\alpha, t) = \Gamma d_\alpha^2 B(\alpha, t). \tag{4.38}$$

Expressing the diffusion coefficient in terms of the Heisenberg time, and relating $\gamma$ from (4.24) to the density of states of the full $N_q Q \times N_q Q$ matrix $\rho = t_{\mathrm{Heis}}/(2\pi)$,

$$\gamma(E) \equiv \tfrac{\pi\sigma}{QN_q}\rho(E), \tag{4.39}$$

we find,

$$\Gamma = 2\epsilon^2 \gamma = \epsilon^2 |V|^2 \frac{t_{\mathrm{Heis}}}{Q}, \tag{4.40}$$

where we introduced $|V|^2 = \frac{\sigma}{N_q}$ following (4.12). This matches the diffusivity found using Fermi's Golden rule in Eq. (2.13). We will see in the next section how this translates into a diffusive behaviour in the spectral form factor.

### 4.3.1 Spectral form factor in perturbed theory

The two-point function of the density of states can be computed from (4.33) using the identity (4.5),

$$\left\langle \rho\left(E + \frac{\omega}{2}\right)\rho\left(E - \frac{\omega}{2}\right)\right\rangle_c = \left(\frac{N_q\gamma}{\sigma}\right)^2 \frac{2}{\pi^2}\mathrm{Re}\left\langle \sum_{n,m} \mathrm{Str}\left[b_n\tilde{b}_n P_b\right]\mathrm{Str}\left[\tilde{b}_m b_m P_b\right]\right\rangle \tag{4.41}$$

$$= \frac{1}{2\pi^2} \text{Re} \sum_{n=0}^{Q-1} \left( \frac{i}{\omega + 2i\delta + i\Gamma_n} \right)^2 \tag{4.42}$$

The SFF is the Fourier transform of the above expression, (4.7),

$$\text{SFF}(t) = \frac{1}{\rho(E)^2} \int d\omega \, e^{-i\omega t} \, \langle \rho(E + \omega/2)\rho(E - \omega/2) \rangle_c$$

$$= \frac{1}{2\pi^2 \rho(E)^2} \, \text{Re} \left[ \sum_n \int d\omega \, e^{-i\omega t} \left( \frac{i}{\omega + 2i\delta + i\Gamma_n} \right)^2 \right] . \tag{4.43}$$

On performing the Fourier transform and taking the limit in which the regulator $\delta \to 0$ vanishes, we get the following expression,

$$\text{SFF}(t) = \frac{|t|}{2\pi \, \rho(E)^2} \Big( \Theta(t) + \Theta(-t) \Big) \times \sum_{n=0}^{Q-1} e^{-\Gamma_n |t|} . \tag{4.44}$$

Note that the first term in the RHS of the above expression is the SFF of an RMT. Therefore, it is only natural to interpret the second term containing the sum over the discrete modes, $n$, as the effective number of symmetry sectors in the theory as a function of time,

$$N_{\text{sectors}}(t) = \sum_{n=0}^{Q-1} e^{-\Gamma_n t} .$$

Using $\Gamma_n = 4\Gamma \sin^2 \frac{\pi n}{Q}$, we estimate $N_{\text{sectors}}(t)$ by converting the sum over discrete modes into an integral as we did in (2.18),

$$N_{\text{sectors}}(t) \approx \frac{Q}{\pi} \int_0^{\pi} dk \, e^{-\Gamma k^2 |t|} = \frac{Q}{\sqrt{4\pi\Gamma|t|}} \, \text{Erf}(\pi\sqrt{\Gamma|t|}) . \tag{4.45}$$

At early times, this function predicts that $N_{\text{sectors}}(t) \approx Q$ as one would expect since the diffusion hasn't washed away the sectors yet. We also get the predicted decay in the intermediate time regime,

$$N_{\text{sectors}}(t) \simeq \frac{Q}{\sqrt{4\pi\Gamma t}} . \tag{4.46}$$

This diffusive behaviour ends when the $N_{\text{sectors}}(t) \simeq 1$, that is, when there is effectively only one sector.[14] At this timescale, $t_f \sim \frac{Q^2}{4\pi\Gamma}$, the SFF has the same behaviour as that of an RMT of the same size as the full Hilbert space,

$$\text{SFF}(t) = \frac{1}{\rho(E)} \frac{t}{t_{\text{Heis}}} \ , \quad t_{\text{Heis}} \geq t \gg t_f \tag{4.47}$$

till it plateaus at Heisenberg time. Thus, we have explicitly derived the diffusive behaviour in the SFF using the $\sigma$-model on the supercoset manifold, $\mathcal{M}_q$, given by (4.27). This EFT is valid for timescales after the Thouless time of the symmetry-preserved theory, $t_{\text{Th},0}$ and is therefore universal for all chaotic theories.

### 4.3.2 Generalisation to varying block sizes

We now generalise the Hilbert space diffusion for a theory where the block sizes of the different charge sectors are unequal, labelled by $N^\alpha$. The total Hilbert space dimension is the sum of dimensions of each of the $Q$ charge sectors, $\sum\limits_{\alpha=1}^{Q} N^\alpha = L$. For such a theory, the appropriate distribution for the Hamiltonian is,

$$P(H_0) = \prod_\alpha e^{-\frac{L}{2\sigma Q} \text{Tr}\left[H_0^{\alpha\alpha}\left(H_0^{\alpha\alpha}\right)^\dagger\right]} \ , \tag{4.48}$$

$$P(V) = \prod_{\alpha<\beta} e^{-\frac{L}{\sigma Q} \text{Tr}\left[V^{\alpha\beta}\left(V^{\alpha\beta}\right)^\dagger\right]} \ . \tag{4.49}$$

This distribution ensures that the interaction strength in the different sectors of the Hamiltonian are the same while facilitating a saddle point analysis. This modified potential subsequently modifies the action that one obtains in terms of the HS fields, $A^\alpha$, is

$$\langle \mathcal{Z}_{(4)}[z] \rangle = \int dA e^{-\frac{L}{2\sigma Q} \sum\limits_\alpha \text{Str}[(A^\alpha)^2 - 2\epsilon^2 A^\alpha A^{\alpha+1}] - \sum\limits_\alpha N^\alpha \text{Str} \ln[z^\alpha + A^\alpha]} \equiv \int dA \, e^{-\frac{L}{Q}S[A]} \ , \tag{4.50}$$

and the saddle point equations of motion,

$$\frac{1}{z^\alpha + A^\alpha} = -\frac{L}{\sigma N^\alpha Q} \left(A^\alpha - \epsilon^2 A^{\alpha+1} - \epsilon^2 A^{\alpha-1}\right) \ . \tag{4.51}$$

---

[14]This can also be viewed as the timescale where our integral approximation of the discrete sum breaks down and finite size effects become important. Note that the discrete sum in (4.44) approaches 1 as $t \to \infty$ because $\Gamma_0 = 0$.

Once again, considering $\omega, \epsilon$ perturbatively and solving the saddle point equation with $\omega = 0 = \epsilon$, we obtain the solutions lying on the coset manifolds $\mathcal{M}_\alpha$,

$$A_0^\alpha = -\frac{1}{2}E\mathbb{1}^{bf} \otimes \mathbb{1}^{ra} + i\gamma_\alpha\tilde{\Lambda}_g^\alpha \quad \text{with} \quad \gamma_\alpha = \sqrt{\frac{\sigma N^\alpha Q}{L} - \frac{E^2}{4}} \ . \tag{4.52}$$

Going through the same technical analysis as before, we obtain the corresponding action in terms of the pion fields,

$$S[B, \tilde{B}] = -\frac{2}{\sigma}\sum_\alpha \left(i\gamma_\alpha(\omega_1 - \omega_2)\text{Str}[B^\alpha\tilde{B}^\alpha] - 2\epsilon^2\gamma_\alpha\gamma_{\alpha+1}\text{Str}[(B^{\alpha+1} - B^\alpha)(\tilde{B}^{\alpha+1} - \tilde{B}^\alpha)]\right) \tag{4.53}$$

$$= -\frac{2}{\sigma}\sum_{n,m}\left[i(\omega_1 - \omega_2)\left(\frac{1}{Q}\sum_\alpha\gamma_\alpha e^{2\pi i\frac{\alpha}{Q}(n-m)}\right)\right.$$
$$\left. -8\epsilon^2\left(\frac{1}{Q}\sum_\alpha\gamma_\alpha\gamma_{\alpha+1}e^{2\pi i\frac{\alpha}{Q}(n-m)}\right)\sin\left(\frac{\pi m}{Q}\right)\sin\left(\frac{\pi n}{Q}\right)e^{i\frac{\pi}{Q}(n-m)}\right]\text{Str}[b_n\tilde{b}_m] \ . \tag{4.54}$$

This leads to a diffusion equation for position dependent diffusion constant,

$$-i(\omega_1 - \omega_2)B^\alpha = 2\epsilon^2\left[\gamma_{\alpha+1}(B^{\alpha+1} - B^\alpha) - \gamma_{\alpha-1}(B^\alpha - B^{\alpha-1})\right] \ . \tag{4.55}$$

Depending on the system at hand, one can make appropriate approximations to solve these equations. Our numerical analysis for the cSYK model suggests that the toy-model with constant block sizes is sufficient to explain the Hilbert space diffusion in the cases where the size of the blocks vary slowly. However, it would be interesting to understand if any interesting new physics might arise in this more general case. It would be particularly interesting to investigate whether very small block sizes coupled with in-homogeneous diffusion constants could cause the system to get stuck in a bottleneck and thus impede the diffusive approach to ergodicity established in this paper.

# 5   Discussion

Let us now take stock of what we have achieved in the present paper before listing some interesting open directions suggested by the results herein. The main result presented in

this paper, backed up by detailed numerical tests and analytic arguments, is a prediction for the Thouless time of quantum many-body systems whose Hilbert space can be gainfully organised by making use of a weakly broken symmetry. In this case, the Thouless physics is controlled by two in principle independent mechanisms. Firstly, the potentially strongly coupled, approach to maximally randomised dynamics in each charge sector, and secondly the exploration of the full Hilbert space of all charge sectors combined, characterised by effectively randomising the off-diagonal blocks of the Hamiltonian. In this paper we have demonstrated that the latter proceeds via a Hilbert-space local diffusion process, which we characterised by a diffusive interpolation visible in the spectral form factor from an early time behaviour indicative of a symmetry-enhanced SFF to a late time non-enhanced SFF indicating quantum ergodic behaviour encompassing the entire Hilbert space. Under the natural assumption that the diffusive process governs the late-time behaviour, we predict the relevant Thouless time when the non-enhanced SFF begins to be given by $Q^2/(4\pi\Gamma)$ (see (1.1)), with $\Gamma$ the Hilbert-space diffusion constant. The underlying mechanism of Hilbert-space local diffusion is analytically confirmed by a supersymmetric sigma model analysis, which explicitly gives rise to a diffusion equation among the sigma-model target spaces associated to each charge sector, controlled by the rate $\Gamma$, as suggested by our heuristic Fermi Golden-Rule analysis in accordance with numerics.

We have also studied the case in which the symmetry is broken by a perturbation that correlates all the charge sectors with each other. This presents a case of 'non-local' Hilbert space diffusion that is exponentially fast. After the onset of non-local diffusion, the effective amount of symmetry of the system at a given time as measured by $N_{\text{sectors}}(t)$ decays exponentially fast until the symmetry is completely washed out, and we are left with an ergodic theory that has explored the entire Hilbert space. These results once again match with the numerical studies of these systems.

We will conclude this article with a discussion of interesting directions, which we hope to address in the future.

## 5.1   Future directions and applications

We would like to further explore the case of inhomogeneous diffusion constant that arises when the individual symmetry-reduced blocks of the Hilbert space do not all have the same dimension. In a similar vein, but of increased technical complication, would be to set up the sigma-model analysis of such more realistic systems from first principles, that is to

explicitly model each Hilbert-space block with the ergodic physics of the SYK model [20,23], rather than a full RMT average as done in Section 4. This would make the interplay of the 'strongly-coupled' Thouless physics of each block with the diffusive approach to a full ergodic limit explicit, and would give valuable insight into the more general problem. A logical next step would then proceed to the incorporation of spatial locality, which we again would expect to be of diffusive nature, but this time in real space. A model incorporating all these aspects would capture all relevant physical mechanism of the approach of the ergodic limit in spatially local many-body quantum systems, including those of local QFTs restricted to a microcanonical energy window and placed on a compact manifold.[15] There, one expects interesting crossovers between the different mechanisms at work, in particular as the Thouless time of an individual sector is expected to be governed by the local number of degrees of freedom $t_N^{\mathrm{Th}} \propto N^\alpha$ (see [18,20,23]) while the two diffusive times are governed by the Hilbert-space diffusion constant $t_{\mathrm{Hilbert}}^{\mathrm{Th}} \propto Q^2/\Gamma$ and real-space diffusion $t_{\mathrm{space}}^{\mathrm{Th}} \propto L^2/D$, where $L$ is the characteristic size of the spatial manifold (the black-hole horizon) and $D$ is the real-space diffusion constant. In the case of real-space diffusion, one also obtains a $\sigma$-model similar to what we find in this work [11, 13, 26]. One particularly interesting class of theories we have in mind here would be holographic field theories, such as the $\mathcal{N} = 4$ SYM theory, whose ergodic phase is dual to the very-late time dynamics of bulk black holes [3, 6]. This competition between various possible Thouless times also applies to local lattice models such as spin chains or unitary circuits, with or without large local Hilbert space dimension. The results of [34], where a Hilbert space that is a tensor product of two independent Hilbert spaces at distinct sites with a weak interaction correlating them induces a diffusive behaviour, might be relevant in such studies.

In this context, it would also be enlightening to reveal the holographically dual mechanism to the process of Hilbert-space diffusion. Given the results of [45], who identify hydrodynamic modes in the spectral form factor [22, 24] in terms of quasinormal modes arising from the so-called double-cone wormhole [4], a natural guess is that the bulk mechanism of Hilbert space diffusion caused by a weakly-broken $U(1)$ symmetry is related to the bulk quasinormal modes associated to a bulk Maxwell field with a small mass on the double cone wormhole. In a related context, it would also be interesting to explore the implications of the Hilbert space diffusion in the holographic setup on the question of presence of global symmetries in gravitational theories [46]. It was shown in this work that the presence of ergodic modes might rule out the presence of a global symmetry in gravitational theories. It will be interesting to

---

[15]The latter two conditions are necessary to ensure that there exists the possibility of an ergodic limit of such theories in the first place.

study the role of diffusion in the mechanism that support this conclusion.

Our results can also be fruitfully applied to random unitary circuits and Floquet systems. A model of mass-deformed SYK model has been studied recently as a model to study transitions from ergodic to localized phases [47, 48]. The absence of conservation laws in these systems allows for our mechanism involving approximate symmetries to control the Thouless time in the thermodynamic limit $L \to \infty$. Many such systems have approximate conservation laws, either by design, or by accident: in particular, disordered systems in 1+1d near a localization transition feature an MBL regime with almost conserved local integrals of motion (see, e.g., [49, 50]). Since the number of conserved charges grows with $L$, our mechanism may explain the observation of Thouless times diverging with system size in these systems [21, 51] (a similar phenomenon is not observed in higher dimensions, consistent with the absence of approximate local integrals of motion there). Depending on details, there may be other explanations for large Thouless times in these systems [17, 21]; however, for systems close enough to an MBL transition, we would expect the long-lived charges to be the bottleneck controlling the approach to RMT.

Another class of models with many approximate conservation laws are systems fine-tuned close to integrability. It would be interesting to study the effective number of sectors $N_{\text{sectors}}(t)$ that enhance the SFF at intermediate times in this context as well. This could be achieved by following an approach similar to [24], by finding a representation of the SFF involving the effective action for generalized hydrodynamics with weak relaxation terms for the approximate charges (see, e.g., [52]).

# Acknowledgments

We would like to thank Alexander Altland, Amos Chan and Moshe Rozali for helpful conversations and comments. This work has been supported in part by the Fonds National Suisse de la Recherche Scientifique (Schweizerischer Nationalfonds zur Förderung der wissenschaftlichen Forschung) through ProjectGrants200020 182513, the NCCR51NF40-141869 The Mathematics of Physics (SwissMAP), and by the DFG Collaborative Research Center (CRC) 183 Project No. 277101999 — project A03. JS thanks Harvard University and the Black Hole Initiative for hospitality during the final stages of preparation.

# Appendix

## A  Perturbation analysis around the saddle point

In this appendix, we study the value of the action (4.20) on the coset manifold $\mathcal{M}_q$, (4.27). In the absence of perturbations, $\omega = 0 = \epsilon$, the action stays constant along this coset manifold courtesy of the $U(1,1|2)^Q$ symmetry. However, when these parameters are switched on, the symmetry explicitly breaks to $U(1|1) \times U(1|1)$ and we obtain a non-zero action on the coset manifold. Let's remind ourselves of the action that we have found in Eq. (4.20)

$$S[A^\alpha] = \tfrac{1}{2\sigma}\mathrm{Str}\left[(A^\alpha)^2 - 2\epsilon^2 A^\alpha A^{\alpha+1}\right] + \mathrm{Strln}[z^\alpha + A^\alpha], \tag{A.1}$$

with the definition of $z^\alpha$ as

$$z^\alpha = \mathbb{1}^{bf} \otimes \left(E\mathbb{1}^{ra} + (\tfrac{\omega}{2} + i\delta)\sigma_3^{ra}\right)^\alpha. \tag{A.2}$$

Also, the solution to the saddle point equation (4.21), with the explicit form found for $\Lambda$, is

$$A_0^\alpha = -\tfrac{1}{2}E\mathbb{1} + i\Lambda^\alpha\gamma(E) = \mathbb{1}^{bf} \otimes \left(-\tfrac{1}{2}E\mathbb{1}^{ra} + i\gamma\sigma_3^{ra}\right)^\alpha, \tag{A.3}$$

where $\mathbb{1} = \mathbb{1}^{bf} \otimes \mathbb{1}^{ra}$.

### Logarithmic term

We redefine $z^\alpha + A^\alpha \equiv K_0^\alpha + K_\omega^\alpha$, where,

$$K_0^\alpha = E\mathbb{1} + A^\alpha, \quad K_\omega^\alpha = \mathbb{1}^{bf} \otimes (\tfrac{\omega}{2} + i\delta)^\alpha \sigma_3^{ra}. \tag{A.4}$$

Here, $A^\alpha$ are the solutions of the saddle point equation that lie on the coset manifold $\mathcal{M}^\alpha$ as described in the discussion following (4.22). Therefore, it follows from (4.21) that,

$$(K_0^\alpha)^{-1} = -\tfrac{1}{\sigma}A^\alpha. \tag{A.5}$$

We briefly drop the $\alpha$ index to avoid notational clutter. These solutions lying on the $\mathcal{M}^\alpha$ can be parametrized by the elements of the coset manifold (4.26), $U \in U(1,1|2)/U(1|1) \times U(1|1)$,

as orbits of the standard saddle point,

$$A = U A_0 (U)^{-1} \equiv A_0 + \delta A \ , \tag{A.6}$$

where, recall that the standard saddle point is given by,

$$A_0^\alpha = -\tfrac{1}{2} E \mathbb{1} + i \Lambda^\alpha \gamma(E) \ . \tag{A.7}$$

Transforming $A_0$ under $U$ amounts to finding out how the $\Lambda$ matrices transform. In order to do that, we rewrite the $U$- matrices as exponential of so called "pion" fields $B, \tilde{B}$,

$$U = e^{-W}, \quad W = -\left(\begin{smallmatrix} 0 & B \\ \tilde{B} & 0 \end{smallmatrix}\right). \tag{A.8}$$

Taking into account terms up to second order in $W$ we find

$$U \Lambda U^{-1} = \Lambda + 2 \Lambda W + 2 \Lambda W^2 + \dots, \tag{A.9}$$

and therefore the fluctuations are given as,

$$\delta A = i \gamma \Lambda + 2 i \gamma \Lambda W + 2 i \gamma \Lambda W^2 + \dots \tag{A.10}$$

Returning to the logarithmic term in the action we expand it in small $\omega$ up to first order,

$$\mathrm{Strln}[K_0^\alpha + K_\omega^\alpha] = \mathrm{Strln}[K_0^\alpha] + \mathrm{Str}[(K_0^\alpha)^{-1} K_\omega^\alpha] + \mathcal{O}(\omega^2) \tag{A.11}$$

Plugging the expressions from (A.5), (A.6) and (A.10) into the above expression and restoring the previously omitted $\alpha$ indices, leads us to

$$\mathrm{Str}[(K_0^\alpha)^{-1} K_\omega] = -\tfrac{1}{\sigma} \mathrm{Str}[A_0^\alpha K_\omega^\alpha] - \tfrac{2 i \gamma}{\sigma} \mathrm{Str}[\Lambda^\alpha W^\alpha K_\omega^\alpha] - \tfrac{2 i \gamma}{\sigma} \mathrm{Str}[\Lambda^\alpha (W^\alpha)^2 K_\omega^\alpha] + \dots \tag{A.12}$$

We remember that the supertrace is defined as $\mathrm{Str}[M] = \mathrm{Tr}[M^{bb}] - \mathrm{Tr}[M^{ff}]$.[16] By evaluating the terms in eq (A.12), we find that the supertrace $\mathrm{Str}[\Lambda W K_\omega]$ vanishes, and we get,

$$
\begin{aligned}
\mathrm{Str}[K_0^{-1} K_\omega] = {} & \tfrac{E}{2\sigma} \mathrm{Str}\left[\left(\begin{smallmatrix} \omega_1 & 0 \\ 0 & \omega_3 \end{smallmatrix}\right) - \left(\begin{smallmatrix} \omega_2 & 0 \\ 0 & \omega_4 \end{smallmatrix}\right)\right] - \tfrac{i\gamma}{\sigma} \mathrm{Str}\left[\left(\begin{smallmatrix} \omega_1 & 0 \\ 0 & \omega_3 \end{smallmatrix}\right) + \left(\begin{smallmatrix} \omega_2 & 0 \\ 0 & \omega_4 \end{smallmatrix}\right)\right] \\
& - \tfrac{2 i \gamma}{\sigma} \mathrm{Str}[B^\alpha \tilde{B}^\alpha \left(\begin{smallmatrix} \omega_1 & 0 \\ 0 & \omega_3 \end{smallmatrix}\right) - \tilde{B}^\alpha B^\alpha \left(\begin{smallmatrix} \omega_2 & 0 \\ 0 & \omega_4 \end{smallmatrix}\right)] + \dots \ .
\end{aligned}
\tag{A.13}
$$

---

[16] $M^{bb}, M^{ff}$ are the components of the graded matrix that map bosons to bosons and fermions to fermions, respectively.

The term $\text{Strln}[K_0^\alpha]$ is invariant under the coset manifold transformations and therefore doesn't expand in fluctuations.

## Quadratic term

Now we study the quadratic term of the action (A.1) and similarly expand it around the saddle point. We immediately see that the term $\text{Str}[(A^\alpha)^2]$ is invariant under $U\alpha$- transformations thanks to cyclicity of the trace. It therefore remains to evaluate

$$-\tfrac{\epsilon^2}{\sigma}\text{Str}\left[A^\alpha A^{\alpha+1}\right]. \tag{A.14}$$

To do so, we define new transformation matrices, $R^{\alpha,\alpha+1}$ as

$$R^{\alpha,\alpha+1} = (U^\alpha)^{-1}U^{\alpha+1}, \quad (R^{\alpha,\alpha+1})^{-1} = (U^{\alpha+1})^{-1}U^\alpha = R^{\alpha+1,\alpha}. \tag{A.15}$$

These measure the relative location on the coset manifold of the saddle point solution at $\alpha+1$ with respect to the solution at $\alpha$. It is straightforward to check that the $R$ matrices can be written in a similar exponential form as $U$ by using

$$V^{\alpha,\alpha+1} = W^\alpha - W^{\alpha+1} \quad \text{and} \quad R^{\alpha,\alpha+1} = e^{V^{\alpha,\alpha+1}}. \tag{A.16}$$

And subsequently we find the transformation properties of $\Lambda$ as (dropping again the $\alpha$ indices for notational convenience)

$$R\Lambda R^{-1} = \Lambda + 2\Lambda V + 2\Lambda V^2 + \dots \tag{A.17}$$

up to second order in the $B$- fields. Plugging in the fluctuations in eq (A.14) we find,

$$\text{Str}[A^\alpha A^{\alpha+1}] = \text{Str}[A_0^\alpha A_0^{\alpha+1}] + 2i\gamma\text{Str}[A_0^\alpha \Lambda^{\alpha+1} V^{\alpha,\alpha+1}] + 2i\gamma\text{Str}[A_0^\alpha \Lambda^{\alpha+1}(V^{\alpha,\alpha+1})^2] + \dots . \tag{A.18}$$

For the same reason as before, the linear term in $V$ vanishes. Equivalently, also the terms proportional to $\text{Str}[A_0 A_0]$ and to $\text{Str}[E\Lambda V^2]$ evaluate to zero.

The non-vanishing contribution is therefore written in terms of the $B, \tilde{B}$ fields,

$$-\tfrac{\epsilon^2}{\sigma}\text{Str}[A^\alpha A^{\alpha+1}] = \tfrac{4\epsilon^2\gamma^2}{\sigma}\text{Str}[(B^{\alpha+1} - B^\alpha)(\tilde{B}^{\alpha+1} - \tilde{B}^\alpha)] + \dots , \tag{A.19}$$

and the total action (A.1) expanded around the saddle point fluctuations is

$$S[A_0^\alpha] = \text{Strln}[E\mathbb{1} + A_0^\alpha] + \tfrac{E}{4\sigma}\text{Str}[\left(\begin{smallmatrix} \omega_1 & 0 \\ 0 & \omega_3 \end{smallmatrix}\right) - \left(\begin{smallmatrix} \omega_2 & 0 \\ 0 & \omega_4 \end{smallmatrix}\right)] - \tfrac{i\gamma}{2\sigma}\text{Str}[\left(\begin{smallmatrix} \omega_1 & 0 \\ 0 & \omega_3 \end{smallmatrix}\right) + \left(\begin{smallmatrix} \omega_2 & 0 \\ 0 & \omega_4 \end{smallmatrix}\right)]$$
$$- \tfrac{2i\gamma}{\sigma}\text{Str}[B^\alpha\tilde{B}^\alpha\left(\begin{smallmatrix} \omega_1 & 0 \\ 0 & \omega_3 \end{smallmatrix}\right) - \tilde{B}^\alpha B^\alpha\left(\begin{smallmatrix} \omega_2 & 0 \\ 0 & \omega_4 \end{smallmatrix}\right)] + \tfrac{4\epsilon^2\gamma^2}{\sigma}\text{Str}[(B^{\alpha+1} - B^\alpha)(\tilde{B}^{\alpha+1} - \tilde{B}^\alpha)] + \ldots \ . \tag{A.20}$$

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
