# Peer review of "Hilbert Space Diffusion in Systems with Approximate Symmetries"

_SciPost Physics_

## Round 4 · Referee Report · Diptarka Das (Referee 2) · 2025-5-2

Strengths
The authors describe universal behaviour of the spectral form factor (SFF) in presence of approximate global symmetries in a generic quantum system.
-
A particular strength of the work is the integration of the analysis into the sigma model description of chaotic systems. The pattern of the symmetry breaking dictates the nature of approach of the SFF ramp towards the random matrix theory ( RMT) predicted ramp. The sigma model effective action analysis via saddle point methods in the limit of large number of sectors ($N_q$ ) justify that the SFF factorizes into the RMT SFF times the time dependent number of sectors that decay from $N_q$ to 1.
-
The authors go on to study various scenarios and also confirm their quantitative predictions via very clear and reproducible numerics.
Weaknesses
Report
The paper resolves a significant open question regarding how approximate symmetries affect the approach to random matrix universality. Prior work established how exact symmetries affect spectral statistics, but the intermediate regime of approximate symmetries remained poorly understood. This work provides a comprehensive framework for this transitional regime. The work has significant implications for understanding quantum thermalization and information scrambling in complex (even dynamical) systems which most of the time have approximate symmetries. I recommend this article for publication.
I also seek a few answers which the authors may choose to answer either as reply to this report / as additional parts to their paper, however some of these questions maybe outside the scope of the present work.
-
Is there a generalization of the $\sqrt{t}$ diffusive behaviour to other power laws when the perturbation connects not all but finite number (>1) of sectors? If there is a hierarchy of symmetry breakings, what kind of generalized diffusion is expected? Are there any such hierarchical SFF behaviour examples that the authors’ work may explain?
-
Can there be any interesting universalities also around the AA saddle that can carry any transport signatures of the approximate symmetries in the plateauing behaviour, are these supposed to always be 1/L suppressed or can they compete when $N_q$ scales as some power of L (for instance in Hilbert space fragmentation like scenarios) ?
-
Do the authors expect that the lessons of approximate symmetry breaking and Hilbert space diffusion also to hold for other scenarios [e.g. they mention U(1) breaking holographic example in 5.1] An interesting scenario is the independent spin sector chaos in a large c conformal field theory (CFT) as discussed in https://arxiv.org/abs/2302.14482. If the spin organization gets broken due to a deformation of the CFT, can one import the results from here to say anything about Hilbert space Diffusion phenomena in the perturbed CFT / can it give any holographic clues ?
-
If the authors had included $\beta$ dependence also in the SFF or discussed a microcanonical filtered SFF, then can the bottleneck mentioned at the end of section 4 be made more quantitative?
Recommendation
Publish (easily meets expectations and criteria for this Journal; among top 50%)
Report
OLD REPORT:
In this paper, the authors study spectral form factor in chaotic system with approximate symmetries. Random matrix universality explains the ramp in the spectral form factor, however any structure in the system leads to deviation from the random matrix result. The authors quantitatively find these deviations from the low lying modes of the symmetry broken effective action. This sigma model analysis closely mirrors chiral symmetry breaking and chiral perturbation theory in QCD.
The authors consider in detail two cases for the symmetry breaking term. 1) Local interactions that connect only neighboring charge sectors which leads to a diffusive $\sqrt{t}$ behaviour for the spectral form factor at intermediate times. 2) All to all, non-local interactions of the charge sectors that leads to a fast exponential decay to the ramp. Both these cases are also verified numerically in a toy model. The first case is also studied in the charged SYK model. In all cases the authors find excellent agreement between numerics and the analytic predictions.
This paper meets all the general acceptance criteria of SciPost Physics and satisfies multiple expectations. For a general system, it is hard to get analytical handle on the spectral form factor, and often numerical analysis is the only viable tool. So the fact that the authors have a quantitative understanding is impressive. Their method is based on symmetry breaking and could be applied to a myriad of systems in the future, from many-body systems to holographic theories. Many potential applications are discussed in the final section.
Overall it is a good piece of work and I recommend it for publication.
I have only one question for the authors. In the case of local interactions, the process is diffusive and the Thouless time is found to be $t_{Th} \sim \frac{Q^2}{\Gamma}$. However when the interaction term is non-local, the Thouless time is not explicitly stated. Is it $t_{Th} \sim \frac{1}{Q \Gamma}$? Or is it harder to estimate?
Recommendation
Publish (meets expectations and criteria for this Journal)

Author: Pranjal Nayak on 2025-12-13 [id 6144]
(in reply to Report 2 by Diptarka Das on 2025-05-02)The comment author discloses that the following generative AI tools have been used in the preparation of this comment:
AI LLM models were used solely to reformulate and polish the response.
We thank the referee for their thoughtful report and the strong recommendation. Below we respond to the specific points raised.
The \(\sqrt{t}\) scaling that we obtain is the universal outcome whenever the perturbation is local in Hilbert space, meaning that it only connects a finite number of nearby charge sectors. If the perturbation is non-local and connects all sectors with roughly equal strength, the decay is instead exponential. Between these two extremes, one can imagine cases with intermediate or long-range couplings across sectors and the resulting exploration of Hilbert space would presumably lead to different power laws, though we have not studied this explicitly.
In cases where the symmetry is broken in a hierarchical fashion, so that some groups of sectors mix quickly while others mix only at longer times, our framework predicts a sequence of diffusion regimes. In practice this would appear as multiple \(\sqrt{t}\) stages, each saturating once the corresponding block of sectors has been explored. The generalized diffusion equation that we present in Sec. 4 is the natural starting point for describing such scenarios.
In our analysis, we have imposed the hierarchy \(t_{\mathrm{Th},0} < t_{\mathrm{diff}} < t_{\mathrm{Heis},0}\), so that the diffusive dynamics are initiated before the original Heisenberg time associated with the unbroken symmetry sectors. Within this window, the diffusive modes that we compute provide a perturbative channel for relaxation that dominates the non-perturbative AA solutions. Consequently, our analysis is controlled by the diffusive modes around the standard saddle and we do not expect visible AA-saddle signatures in this regime. Moreover, for the observables in question, the agreement between the numerics and the analytics is strong enough to substantiate this intuition.
It is nevertheless plausible that there might be other scenarios—for example, when the number of sectors grows with the system size or the sector coupling is extremely weak—where the crossover to the AA regime could overlap with the tail of the diffusive dynamics. In such situations, one might expect contributions from pseudo-AA saddles, configurations that resemble the AA saddle of the unperturbed problem but are no longer exact saddle points once the sectors interact. Quantifying such effects would require going beyond the scope of the present work, though it is a well-posed and interesting direction for future study.
We thank the referee for highlighting Ref. [2302.14482], where it is discussed that in large-c chaotic CFTs, each spin sector can exhibit its own random-matrix universality. Our approach should help understand how weak violations of spin conservation could couple these sectors. The spin label would play the role of an approximate charge, and small spin-changing deformations would induce diffusion in this sector-space, leading to the exploration of sector space and to the build-up of the final RMT ramp. In principle, our framework should also be able to model this scenario.
A possible holographic setup to explore is a weakly broken \(U(1)\) symmetry that manifests as a light bulk gauge field with a small mass and whose quasinormal-mode decay rate sets the decay rate \(\Gamma\). The resulting diffusion process would then correspond to the decay of this bulk mode. It would be interesting to explore this connection more concretely on the double-cone wormhole background, where such quasinormal modes could provide the bulk counterpart of the diffusive dynamics described in our work.
In Sec. 4 we pointed out a plausible bottleneck in the dynamics, namely that the approach to full random-matrix behaviour can be slowed down in systems with very small block sizes and/or inhomogeneous diffusion constants, because diffusion in Hilbert space has to complete before the universal regime can set in.
By including an energy filter (microcanonical or thermal), the diffusion constant \(\Gamma\) would become energy dependent, resulting in an energy-dependent diffusion time, the bottleneck time. In principle, our formalism is able to capture such effects and it would be interesting to do so in the future. Also, in the numerics for SYK in Figure 4, we already noticed that the spectrum near the edge plays a role in visibly smoothing the transition to the plateau, and we therefore made use of a Gaussian filter to suppress contributions from eigenvalues near the edge.
We thank the referee again for these insightful suggestions.
With kind regards,
The authors

---

## Editorial Decision

unknown